



# Indian ocean marine biogeochemical variability and its feedback on simulated South Asia climate

Dmitry V. Sein[1,2], Anton Y. Dvornikov[1], Stanislav D. Martyanov[1], William Cabos[3], Vladimir A. Ryabchenko[1], Matthias Gröger[4], Daniela Jacob[5], Alok Kumar Mishra[6], Pankaj Kumar[6]

[1]Shirshov Institute of Oceanology, Russian Academy of Sciences; Moscow, Russia
[2]Alfred Wegener Institute, Helmholtz Centre for Polar and Marine Research; Bremerhaven, Germany
[3]University of Alcala; Alcala, Spain
[4]Leibniz Institute for Baltic Sea Research, Warnemünde, Rostock 18119, Germany
[5]Climate Service Center Germany (GERICS); Hamburg, Germany
[6]Department of Earth and Environmental Sciences, Indian Institute of Science Education and Research Bhopal, India

*Correspondence to*: Dmitry V. Sein (dmitry.sein@awi.de)

**Abstract.** We investigate the effect of variable marine biogeochemical light absorption on Indian Ocean sea surface temperature (SST) and how this affects the South Asian climate. In twin experiments with a regional Earth System Model, we found that the average SST is lower over most of the domain when variable marine biogeochemical light absorption is taken

into account, compared to the reference experiment with a constant light attenuation coefficient equal to 0.06 m$^{-1}$. The most significant deviations (more than 1°C) in SST are observed in the summer period. A considerable cooling of subsurface layers occurs, and the thermocline shifts upward in the experiment with the activated biogeochemical impact. Also, the phytoplankton primary production becomes higher, especially during periods of winter and summer phytoplankton blooms. The effect of altered SST variability on climate was investigated by coupling the ocean models to a regional atmosphere model. We find the

largest effects on the amount of precipitation, particularly during the monsoon season. In the Arabian Sea, the reduction of the transport of humidity across the equator leads to a reduction of the large-scale precipitation in the eastern part of the basin, reinforcing the reduction of the convective precipitation. In the Bay of Bengal, it increases the large-scale precipitation, countering convective precipitation decline. Thus, the key impacts of including the full biogeochemical coupling with corresponding light attenuation, which in turn depends on variable chlorophyll-a concentration, include the enhanced

phytoplankton primary production, a shallower thermocline, decreased SST and water temperature in subsurface layers, with cascading effects upon the model ocean physics which further translates into altered atmosphere dynamics.

## 1 Introduction

The vulnerability and the ability of society and natural systems to adapt to the impact of climate change vary significantly according to geographic regions and populations. The Indian subcontinent and adjacent area, where a fifth of humanity lives,

is one of the regions where the impacts are substantial both in the present time and future climate projections (Turco et al., 2015; Szabo et al., 2016). The strongest impacts are related to changes in the intensity and frequency of extreme events, such



as floods, droughts, tropical cyclones, storm surges, phytoplankton blooms, ocean heat waves, avalanches, etc. which can inflict significant damage on ecosystems, human populations, infrastructure and property (IPCC AR5, 2014).

Atmospheric extreme events contribute to the emergence of extreme situations in the ocean and vice versa. For example, the strengthening of the southwestern monsoon in the Arabian Sea leads to abnormal coastal upwelling. It increased mixing of the upper ocean layer, and the subsequent supply of nutrients into the upper layer from the deep ocean favors anomalous blooms of phytoplankton (Ryabchenko et al., 1998). In turn, the changes in sea surface temperature (SST) and surface fluxes of heat and momentum caused by monsoons can feedback to atmospheric circulation. Another example of the relationship between atmospheric and oceanic processes is associated with river runoff and nutrient loading which is projected to be maximum in southern and eastern Asia due to population growth and increased industrialization (Seitzinger et al., 2002). It was stated that the estuarine ecosystem experiences a complete change in terms of phytoplankton during monsoon (De et al., 2011) and that the eastern Indian coast is affected by localized eutrophication which directly influences the nutrient level of coastal water and phytoplankton abundance (Choudhury & Pal, 2010). Recent assessments (Sattar et al., 2014) of the impact of food production upon the river flux of nutrients into the Bay of Bengal coastal waters in the past and the future show that the coastal eutrophication potential is high in the Bay of Bengal, thus elevating the risk for oxygen deficiencies (D'Asaro et al., 2019). The above examples of interactions between atmospheric and oceanic processes underscore the need to create a unified high-resolution modelling system for the region to be able to study these interactions in detail.

Earth System Models (ESMs) are very effective tools for the study of complex systems and associated mechanisms in climate and environmental sciences in the past and future, driven by assumptions on the evolution of climate change (Taylor et al., 2012). However, they usually lack the resolutions that are necessary for regional studies. Regional Climate Models (RCMs) are used to translate the global climate information generated by ESMs down to regional scales at a higher resolution. RCMs take the initial conditions and time-dependent boundary conditions from the global models and provide dynamically downscaled climate information within the region of interest (Giorgi, 2006).

We have implemented a new version of the high-resolution Regional Earth System Model (RESM) ROM (Sein et al., 2015) for South Asia and the northern Indian Ocean. The model includes ocean, atmosphere, hydrological cycle and marine biogeochemistry components. Such a modelling system is required for the study of extreme events in the atmosphere and the ocean in the India region, for seasonal and decadal predictions, climate change projections and advanced monsoons modelling. In this study, we will use the model to assess the impact of a fully coupled interactive marine biogeochemical model upon the simulation of the present climate over the Indian subcontinent and the adjacent ocean using the South Asia CORDEX domain (CORDEX - Coordinated Regional Climate Downscaling Experiment, https://cordex.org/domains/region-6-south-asia-2). Monsoon dynamics are sensitive to changes in SSTs and so a model representing the Indian Ocean should involve all relevant processes to control the heat budget of the near surface ocean. We here focus on the impact of variable chlorophyll concentration on SSTs and how this feeds back on climate. A number of studies focused on the investigation of the marine biogeochemistry impact upon physical properties of the ocean showed that the general scheme is as follows: the presence of phytoplankton leads to the warming of the ocean upper layer and the cooling of subsurface layers (e.g., Nakamoto et al., 2000;



Lengaigne et al., 2007; Park et al., 2014a; Park et al., 2014b). Nevertheless, in some circumstances, the presence of phytoplankton, as reported by (Nakamoto et al., 2001; Manizza et al., 2005; Park et al., 2014b), may lead to the cooling of the surface layer as well due to enhanced upwelling of cold subsurface water in the eastern equatorial Pacific. Still, the warming of the eastern equatorial Pacific due to the influence of biological productivity was reported by (Lengaigne et al., 2007), who

compared fully coupled ocean-atmosphere-biogeochemistry model experiment with fixed-chlorophyll model experiment. They also discussed the inconsistency between the results of forced ocean models and the fully coupled models, suggesting that the impact of marine biogeochemistry upon SST and corresponding cooling/warming is related to the way radiation is treated in the control experiments. The climate variability in the Indian Ocean was studied by (Park & Kug, 2012), who also showed that the presence of chlorophyll increases the mean SST due to biological heating. Therefore, a significant number of

published studies on biogeochemical influence upon ocean physics have shown that taking into account phytoplankton's presence while computing the penetration of the short-wave solar radiation into the water leads to the warming of the surface layer and cooling of subsurface layers of the ocean. One of the main approaches in those studies was setting the phytoplankton concentration equal to zero (a reference experiment, so-called "dead ocean"), to a constant value (studying the influence of phytoplankton upon ocean physics, but not vice versa), or using a fully interactive simulated phytoplankton concentration

affecting the SWR absorption when all possible feedbacks between physical and biogeochemical models are taken into account (e.g., Manizza et al., 2005; Lengaigne et al., 2007). Our current research differs in that we do not investigate the influence of phytoplankton upon ocean physics in general but investigate how the spatial and temporal variability of marine biogeochemistry affects the regional climate. Given the changing climate reality, such a question seems reasonable and worth discussion. Should the variability of marine biogeochemistry be taken into account when performing ESM climatic

predictions? Which part of the climatic system will be affected, and which will not?

To answer these questions, we compare two simulations carried out with our RESM. These two simulations differ only in the influence of the ocean biochemistry module on the shortwave solar radiation penetration into the ocean. In the first simulation, we use a constant in time and space light attenuation coefficient corresponding to Jerlov IB water type (Jerlov, 1976; Paulson & Simpson, 1977). Although such a value of the attenuation coefficient may implicitly include the impact of phytoplankton,

it absolutely neglects its spatial and temporal variability. In the second simulation, we introduce the full spatial and temporal variability of marine biogeochemical feedback by calculating the attenuation coefficient using the phytoplankton concentration simulated by the ocean biogeochemistry module following (Gröger et al., 2013).

It is worth mentioning that many previous studies investigating the role of phytoplankton feedback on climate were based mainly on global models, i.e. either global ocean-marine biogeochemistry models or global coupled atmosphere-ocean-marine

biogeochemistry models. These models do not well account for small scale dynamics and thus often yield a spatially smoothed picture of phytoplankton dynamics. In this study, both the atmospheric as well as the oceanic components are in a resolution that fully provides the added value of regionalization compared to global models. This has been demonstrated previously for the North Atlantic as well as the NW European shelf seas (Sein et al., 2015; Sein et al., 2020).



Finally, it is well known that ongoing climate change will reduce the mixed layer depth in many ocean regions together with

a decline of ocean productivity (Steinacher et al., 2010; Fu et al., 2016). Furthermore, changes in atmospheric nutrient depositions turn out as a driver of biological productivity in many oligotrophic regions (Myriokefalitakis et al., 2020). The changes in production are likely to alter the water chlorophyll concentration of the upper ocean and thus on SST. The impacts can not be simulated with models using Jerlov type light attenuation and so the role of phytoplankton in climate change projections is more or less unexplored.

The objectives of this paper can be summed as follows:

1. To evaluate the ability of our model to reproduce the present climate in the South Asia CORDEX region both in the ocean and the atmosphere.

2. To evaluate the quality of corresponding simulated physical and biogeochemical characteristics in the northern part of the Indian Ocean.

3. To assess the impact of the full spatial and temporal variability of marine biogeochemistry's feedback upon the simulated regional climate, both in the atmosphere and the ocean.

The layout of the present paper is as follows. In section 2 a description of the coupled modelling system is presented. Section 3 is focused on the verification of the developed RESM Section 4 contains some discussion. Conclusions are presented in section 5.

## 2 Methods

The oceanic component of ROM is the Max Planck Institute Ocean Model (MPIOM: Marsland et al., 2002; Jungclaus et al., 2013), which is coupled to the REgional atmospheric MOdel (REMO: Jacob, 2001a, 2001b) Via the OASIS coupler. ROM also includes as modules the Hamburg Ocean Carbon Cycle model (HAMOCC: Ilyina et al., 2013), and the Hydrological Discharge model (HD: Hagemann and Dumenil, 1998). MPIOM provides the possibility to refine the grid resolution in the

region of interest and to avoid the lateral boundary conditions in the ocean while performing calculations. Another feature of the ROM system is that the coupling between the ocean and the atmosphere is implemented only at the chosen subdomain. At the same time, outside this region, MPIOM calculates heat, freshwater, and momentum fluxes from atmospheric fields taken from the same global model used for REMO boundary conditions. A detailed description of ROM can be found in (Sein et al., 2015).

In this work, we use for REMO the slightly enlarged South Asia CORDEX domain (http://www.cordex.org), while for MPIOM the global mesh has a variable horizontal resolution which reaches up to 15 km inside the coupled region (Fig. 1). The model is driven by data from a CMIP5 20th century simulation with the MPI-ESM LR setup.



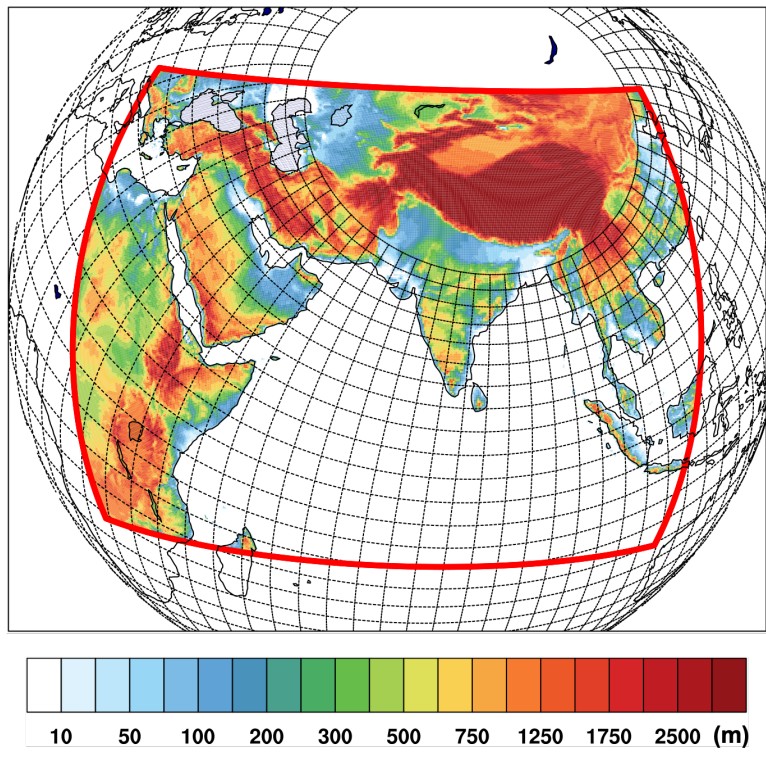

**Figure 1: ROM configuration. The red frame shows the coupled ocean-atmosphere CORDEX domain. The black lines indicate the grid of the MPIOM/HAMOCC models (only every 12th line is shown). Color scale represents orography.**


For this study, we perform two present-time simulations using ROM. The two simulations (labeled as INDJ and INDB hereafter) are almost identical and differ only in the parameterization of the attenuation of shortwave radiation (SWR)

penetrating into the ocean. In the INDJ experiment, we use a constant in time and space light attenuation coefficient equal to 0.06 m$^{-1}$ which corresponds to Jerlov IB water type (Jerlov, 1976; Paulson & Simpson, 1977). Although this parameterization implicitly includes the impact of phytoplankton, it neglects its spatial and temporal variability and has several shortcomings. Firstly, the dynamics of phytoplankton blooms on the light climate is completely neglected, which is highly problematic in regions which are subject to a strong seasonal cycle and in regions with strongly varying nutrient supply. Secondly, coastal

characteristics, especially in front of large rivers with high nutrient load and limited exchange with the open ocean, are not resolved which is however crucial in high resolution downscaling simulations. This experiment was performed for the period 1920–2005, the first 30 years being an adjustment period. Initial conditions for the biogeochemical module were taken from MPIOM/HAMOCC long term simulations (Gröger et al., 2013). For the ocean and atmosphere, the initial conditions were taken from previous spin-up simulations: 50 years MPIOM stand-alone run plus 2 times 40 years coupled MPIOM/REMO

simulations with ERA-Interim forcing.



In the second simulation (INDB), which starts from the beginning of the year 1950 of the INDJ experiment, we introduce the full spatial and temporal variability of marine biogeochemical feedback by calculating the attenuation coefficient using the phytoplankton concentration simulated by the ocean biogeochemistry module as proposed by (Gröger et al., 2013). Hence, the presence of a strong local phytoplankton bloom in the surface layer will increase the heat absorption in the upper layers and decrease it in deeper layers compared to a no-bloom period, with cascading feedback on the thermohaline structure of the water column and heat flux between the ocean and the atmosphere. Due to these reasons, the effect of seasonal and local varying phytoplankton concentration can be expected to be important in regional climate studies. Including the full variability of the marine biogeochemical feedback, as done here, is not a common practice in climate simulations, as it requires online coupling to a biogeochemistry model which leads to a threefold consumption of CPU hours compared to an uncoupled model running with Jerlov water types.

Apart from the physical feedback of phytoplankton on SST and successive ocean-atmosphere heat exchange, the production of phytoplankton lowers the local concentration of dissolved inorganic carbon and thus the pCO2 of the surface water. As a result, the air-sea pCO2 gradient is altered which in turn feeds back on the air-sea carbon exchange. However, as in the overwhelming part of coupled ocean-biogeochemistry-atmosphere models, the air-sea carbon fluxes are passively coupled. Consequently, an increased air to sea carbon flux due to a strong phytoplankton bloom is not communicated to the atmosphere. Consequently, unlike water pCO2, atmospheric pCO2 does not change but is prescribed during the whole simulation. In conclusion, the only way phytoplankton influences the atmosphere is by its impact on SST and subsequent heat fluxes.

## 3 Results

### 3.1 Ocean

For validation of the model results, we use the temperature, salinity, dissolved nitrates and dissolved phosphates data from the World Ocean Atlas 2013 (WOA13), and chlorophyll concentration from the satellite data (SeaWiFS and MODIS-Aqua).

According to the India Meteorological Department, we distinguish the following seasonal periods used for the verification procedure based on the monsoon activity in the northern part of the Indian Ocean:

• DJF: December–February (winter season, NE winds);

• MAM: March–May (pre-monsoon season);

• JJAS: June–September (monsoon season, SW winds);

• ON: October-November (post-monsoon season);

In the following, we compare the model results and observations for winter (DJF) and summer (JJAS) seasons time-averaged over 1975–2004, since the phytoplankton impact is expected to be maximal during the bloom periods.



### 3.1.1 Sea surface

*Sea surface temperature and salinity (SST and SSS).* Figure 2 shows the spatial distribution of the difference between the INDJ and WOA13 SST (Locarnini et al., 2013) and SSS (Zweng et al., 2013) for winter (DJF) and summer (JJAS) averaged over the 1975–2004 period. The model generally underestimates the SST, the exception being the region located off the coast of the Somali peninsula. The most considerable deviations in SSS from the WOA13 are observed in the Bay of Bengal, with overestimations by the model in the 0.5–2‰ range. The largest discrepancy in SSS occurs in winter (DJF), while in pre-monsoon and monsoon seasons, the maximum difference is about 0.5‰ and 1.5‰, respectively. Off the western Indian coast, INDJ shows a somewhat lower SSS than that in the WOA13, with the largest discrepancies occurring in autumn and being up to 1‰ (not shown).

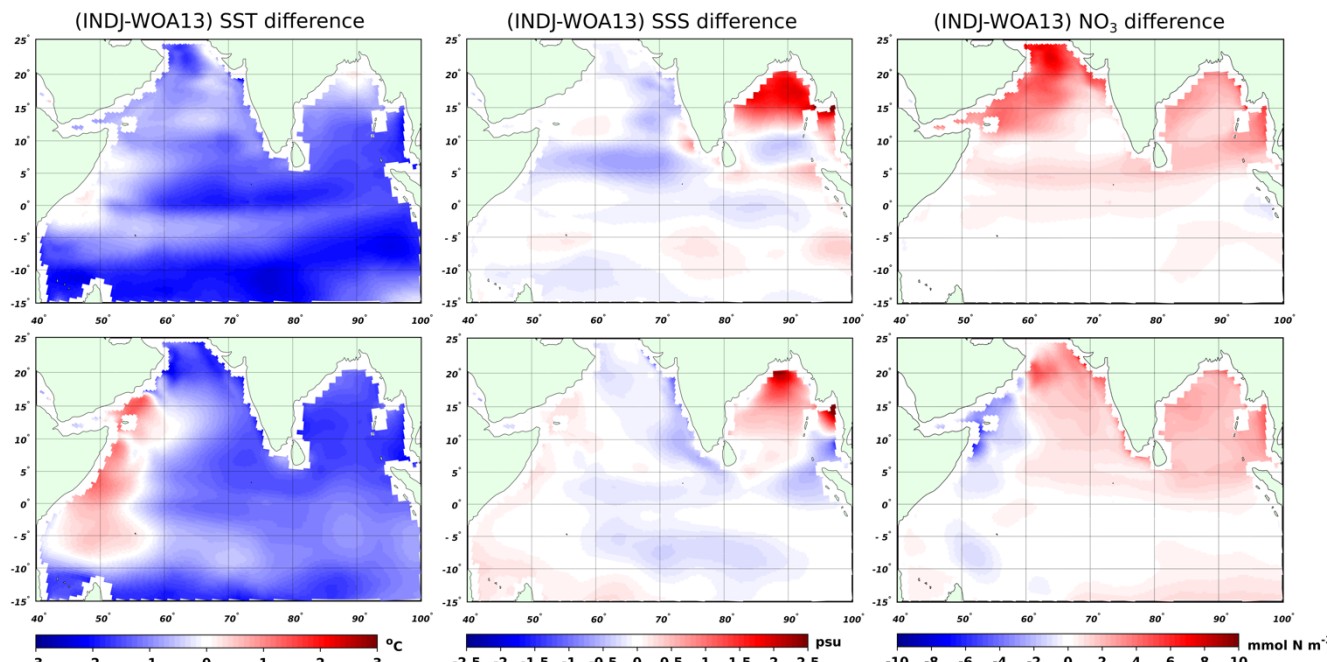

**Figure 2: Spatial distribution of the difference between experiment INDJ and WOA13 for SST (left column), SSS (middle column) and NO₃ in the surface layer (right column). SST, SSS and NO₃ are time-averaged for winter (DJF; upper row) and summer (JJAS; lower row) for the period 1975–2004.**

Sea surface concentration of dissolved nitrate. HAMOCC somewhat overestimates the surface concentration of nitrates (NO3), especially during winter (Fig. 2). The strongest deviations are located along the coasts. They are related to uncertainties in nutrient supply originating from rivers and point sources as we apply a rough climatological estimate for external nutrient supply (Gröger et al., 2013). Further from the coasts, the model biases reduce showing the model's capability to correctly



simulate the biogeochemical cycling of the open Indian ocean which is the primary purpose of this study. Too high SSTs and
too low nitrate concentrations near the NE Africa and South Arabia coast during the summer monsoon season may indicate
that the model produces a too weak upwelling in response to the predominant SW wind regime. The agreement between
WOA13 (Garcia et al., 2013) and the model varies with depth: at 50 m the main features of the spatial distribution of nitrates
are reproduced correctly. The only serious exception is the overestimation of the concentration of nitrates in autumn off the
southwest coast of India. At a depth of about 100 m the discrepancies become more pronounced, while at 500 m the WOA13
and modelled nitrates are very similar, as the influence of the seasonal ecosystem dynamics upon the distribution of nitrates at
such depths becomes small. The maximum deviations in surface nitrate field between the model and WOA13 data occur during
the bloom periods (winter and summer) in both simulations (INDJ and INDB), while this deviation is minimal in spring. In
general, the modelled annual surface concentration of dissolved nitrate is slightly higher than in WOA13.

*Sea surface chlorophyll-a concentration.* For the validation of the ocean surface chlorophyll-a concentration, the surface
phytoplankton concentration (in carbon units) calculated by HAMOCC was converted into chlorophyll-a concentration (in
mg/m3) using a constant C:Chl ratio equal to 60 gC / gChl (Ilyina et al., 2013). Figure 3 demonstrates the spatial distribution
of modelled (INDJ) and observed (SeaWiFS) surface chlorophyll-a concentration for winter and summer climatic seasons.

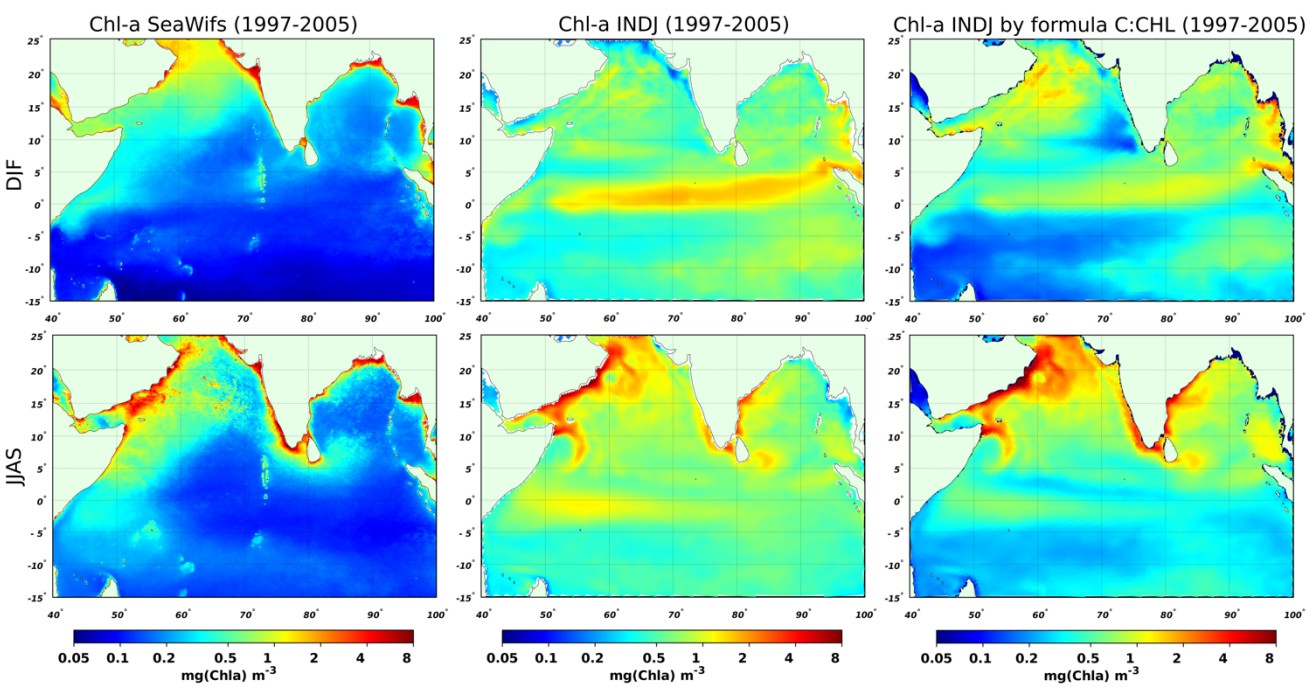

**Figure 3: Distribution of surface chlorophyll-a concentration obtained by the SeaWiFS satellite data (left column),
model using a fixed C:Chl ratio (middle column), and using a varying C:Chl ratio as in (Anderson et al., 2007; right
column), for winter (upper row) and summer (bottom row) climatic seasons.**



It is clear that ROM overestimates the chlorophyll-a concentration in comparison with SeaWiFS satellite data (SeaWiFS,
2020). The model produces lower chlorophyll-a concentrations in the Arabian Sea under the predominant NE wind regime
during the winter monsoon. By contrast, SW winds during the summer monsoon induce the upwelling of nutrients from deeper
layers and stimulate primary production. In winter the model simulates enhanced chlorophyll-a concentrations along the
eastern boundary of the Bay of Bengal while showing their reduced production during summer. While these changes are in
accordance with the changing wind regime, the satellite data also shows higher concentrations during summer. The most
plausible explanation for this is a persistently high supply of riverine nutrients around the year. Another difference between
the model and satellite data is the presence of increased chlorophyll-a concentration zone stretching along the equator in the
model results, especially during the winter period which is not present in satellite data. A good agreement, both qualitative and
quantitative, between model's winds and ERA5's winds (Fig. 17, see below), suggests that the enhanced model's equatorial
surface phytoplankton concentration cannot simply be related to incorrect wind simulation. The problem may be related to
relatively coarse vertical resolution of MPIOM in the upper layer (16 m) together with simple turbulence closure scheme in
MPIOM based on (Pacanowski and Philander, 1981). The overestimation or underestimation of ocean productivity along the
equatorial divergence zone is a common problem of many ocean general circulation models (e.g., Steinacher et al., 2010). (Liu
et al., 2013) also reported and discussed significant discrepancies between observed and modeled chlorophyll-a surface
concentrations in the equatorial Indian Ocean in an ensemble of five CMIP5 coupled models. Their analysis showed that all
the considered models shared the same structures and deviations in that region. Unfortunately, our RESM also has the same
drawback in this really challenging problem.

The overestimation of chlorophyll-a concentration in the domain may also be explained by a relatively simple description of
phytoplankton dynamics in the HAMOCC model. HAMOCC includes only one type of phytoplankton and, as a component of
a global climatic model, it was configured to produce realistic global-mean primary production (Ilyina et al., 2013), but may
significantly over- or underestimate some regional features of marine biological productivity. We suppose that this is the main
cause of differences between satellite and model results. This also holds for overestimated regional concentration of dissolved
nitrate, which is another issue of HAMOCC and other global models (Ilyina et al., 2013).

Another cause of modeled chlorophyll-a overestimation may be the fixed phytoplankton C:Chl ratio used in the HAMOCC
model. As was mentioned above, HAMOCC uses a constant C:Chl ratio equal to 60 gC / gChl, and this ratio is used herein to
convert the modeled surface phytoplankton concentration (expressed in carbon units) into surface chlorophyll-a concentration
(expressed in mg Chl / m3) in order to validate the model's results against SeaWiFS satellite data. Nevertheless, as was shown
in numerous studies, the phytoplankton C:Chl ratio is very variable, depending on specific phytoplankton species, irradiance
level, and bloom phase. The values of C:Chl ratio may be 20-50 at low irradiances and up to 100-200 at high irradiances, as
reported by (Smith & Sakshaug, 1990).

Besides a fixed C:Chl ratio, a functional dependency of C:Chl can be used in some models. For example, in (Anderson et al.,
2007) such function was used in a biogeochemical model of the Arabian Sea, which includes water temperature and nutrient
concentrations as arguments. Figure 3 shows the surface chlorophyll-a concentration (converted from modeled phytoplankton





concentration) calculated with a fixed and variable (Anderson et al., 2007) C:Chl ratio in order to compare it with satellite data and check if a variable phytoplankton C:Chl ratio may give a better agreement with SeaWiFS observations.

As seen from Fig. 3, using the above-mentioned parameterization for phytoplankton variable C:Chl ratio gives better agreement between model and SeaWiFS surface chlorophyll-a concentration. It should be noted that the constant C:Chl ratio is used in HAMOCC in the photoadaptation process, so, strictly speaking, it is not consistent to use another C:Chl ratio for converting the modeled phytoplankton concentration into chlorophyll-a concentration, but still is useful to demonstrate the impact of phytoplankton C:Chl ratio variability upon model's verification.

A comparison of the time-series of the HAMOCC surface chlorophyll-a concentration with satellite data was also carried out (Fig. 4). The general overestimation of simulated chlorophyll-a surface concentrations mentioned above is also evident in these time-series. However, during several short periods the MODIS's daily-mean climatic concentrations (MODIS, 2020) appear to be higher than in the model.




**Figure 4: Comparison of the simulated (INDJ and INDB) and observed (SeaWiFS, MODIS Terra) time-series of surface chlorophyll-a concentration in the Arabian Sea (a), Somali upwelling area (b), and the Bay of Bengal (c). DC and MC - daily-mean and monthly-mean climatic averaging of satellite data for the period 1997-2005, respectively.**



### 3.1.2 Vertical distributions

We have also analysed the spatially averaged vertical profiles of water temperature, salinity, dissolved nitrate and phosphorus concentration for the northern part of the Indian Ocean (IO) and for the Arabian Sea (ASF) and the Bay of Bengal (BBF) regions (Fig. 5).

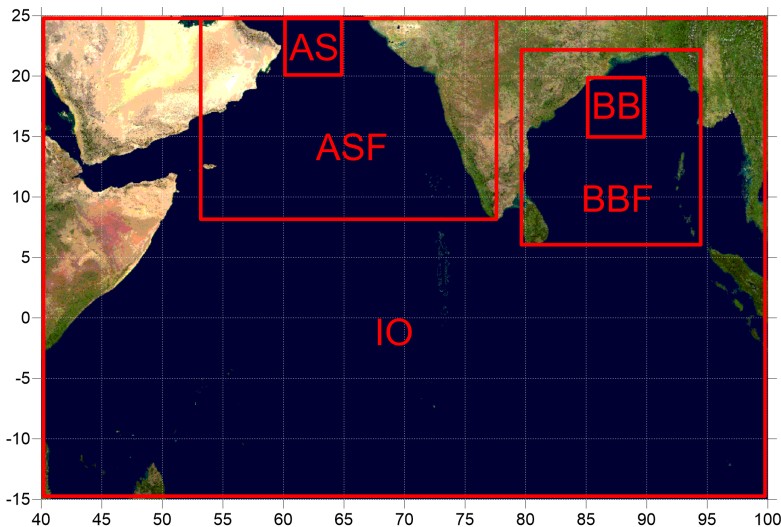

**Figure 5: Regions over which spatial average is computed for the vertical profiles of fields (temperature, salinity and nutrients). Acronyms as used for further analysis: IO - Indian Ocean, ASF - Arabian Sea Full, AS - Arabian Sea box, BBF - Bay of Bengal Full, BB - Bay of Bengal box.**

As seen from Fig. 6, the simulated vertical distribution of temperature and salinity is in relatively good agreement with WOA13

data. Model results are generally within the standard deviation range of the corresponding WOA13 data in ASF and in IO. However, in BBF the modeled temperature and salinity are out of the standard deviation range. Still, it should be noted that the standard deviation of WOA13 temperature in the whole water column and salinity below 100 meters is very small in these areas due to the scarcity of observations. The same is true for the vertical distribution of nutrients (Fig. 7).





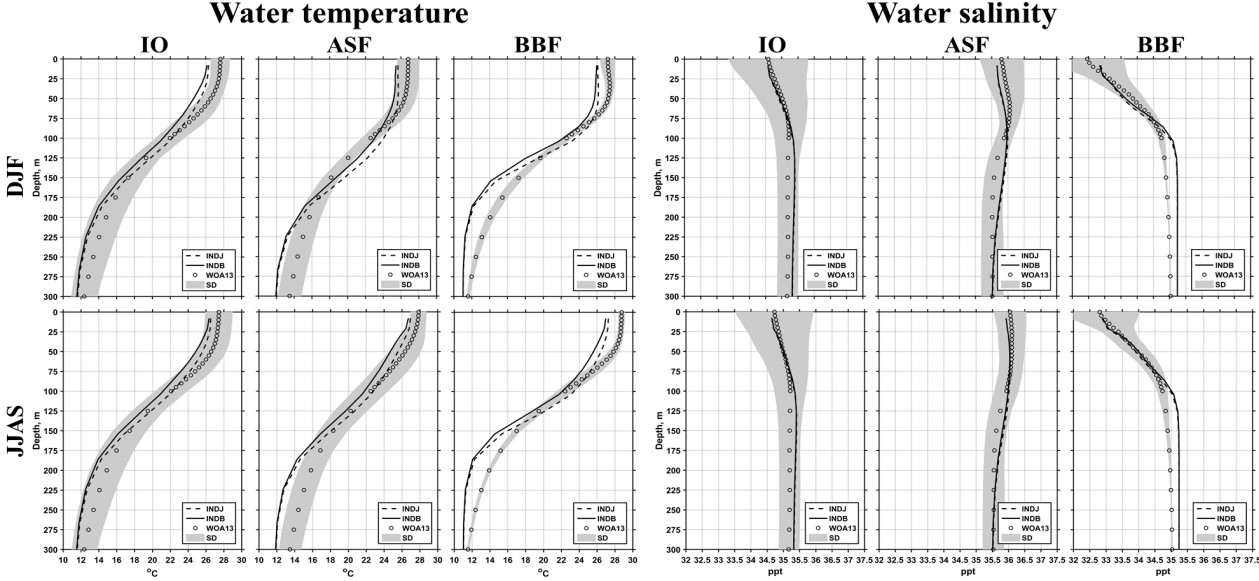

**Figure 6: Vertical profiles of water temperature and salinity time-averaged seasonally (DJF, JJAS) for the period 1975–2004. INDJ and INDB designate the model runs; WOA13 designates the climatic data from the World Ocean Atlas 2013; SD designates the standard deviation of the WOA13 data.**

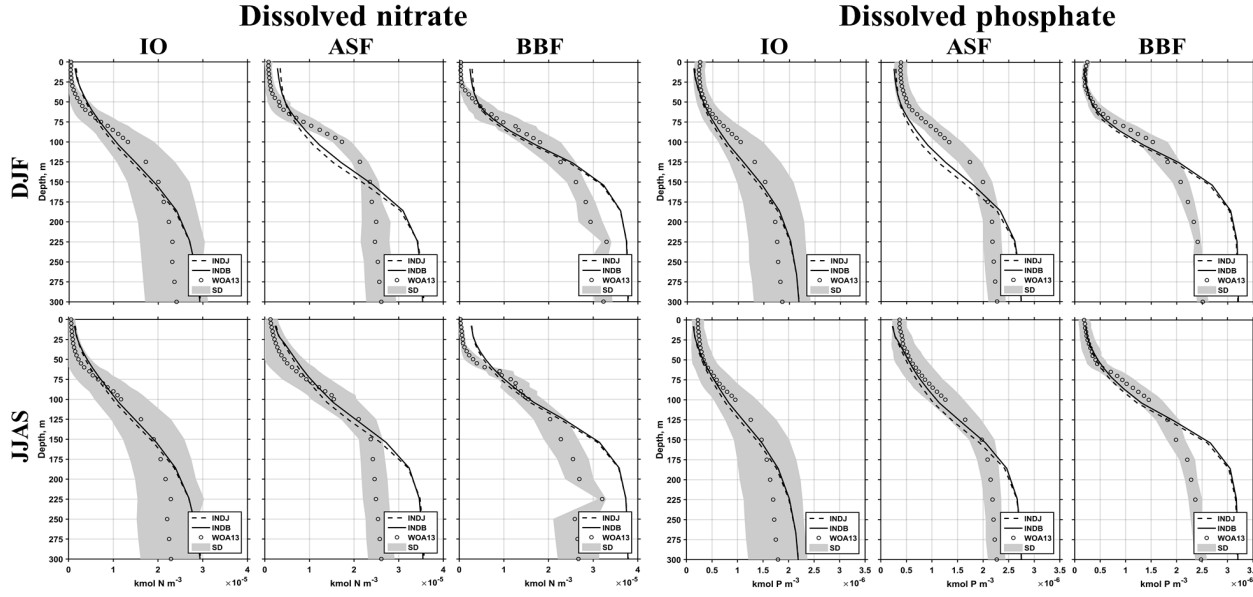

**Figure 7: Vertical profiles of dissolved nitrate and phosphate time-averaged seasonally (DJF, JJAS) for the period 1975–2004. INDJ and INDB designate the model runs; WOA13 designates the climatic data from the World Ocean Atlas 2013; SD designates the standard deviation of the WOA13 data.**



### 3.1.3 Mixed layer depth

A possible way to analyse the impact of biology on the water column is how it affects mixing. Hence, we calculated the mixed layer depth (MLD) according to the 0.2°K criterion and compared it directly to the observational-based mixed layer depth climatology provided by (de Boyer Montégut al., 2004) (Fig. 8). During the SW monsoon season (JJAS) ROM simulates a deeper MLD in a narrow band along the coast of Somali compared with observational data (Fig. 8a, upper panel). Due to higher horizontal resolution of the ocean module MPIOM (up to 15 km), ROM can generally better reproduce small-scale

structures compared to data of (de Boyer Montégut al., 2004) which has horizontal resolution of only 2° and where small-scale structures may be less pronounced. A big mismatch is seen in the eastern Bay of Bengal where INDJ simulates a very deep (>70 m) MLD which is not seen in the observations. This points to a systematic overestimation of the MLD in this area. However, the difference between INDB-INDJ (Fig. 8b, left) shows that this bias is substantially reduced when ROM takes into account the explicit heat absorption by modelled phytoplankton.

During boreal summer (Fig. 8a, JJAS) the MLD deepens in the southern part of the domain in both ROM and observations. However, in the observation, the zone of deep mixing expands more northward compared to ROM.





**Figure 8: a) Upper panel: Summer (JJAS) mixed layer depth as calculated in the INDJ experiment (left) and from**
**305    observational data (de Boyer Montégut al., 2004; right). Lower panel: same as upper panel but for the winter period**
**(DJF). b) Mixed layer depth difference between two model simulations (INDB-INDJ) for summer (left) and winter**
**(right) periods. All fields are time-averaged seasonally (DJF, JJAS) for the period 1975–2004.**

During boreal winter (DJF), the observational data set shows much lower spatial variation of the MLD than in the INDJ
experiment (Fig. 8a, lower panel). Partly, this is expected due to the coarse resolution of the observational data set. However,
it is obvious that in both the Arabian Sea as well as in the Bay of Bengal, ROM seems to overestimate the MLD, whereas in
the southern part of the domain, where the MLD is generally shallower, the differences are less pronounced. In the northern





Indian Ocean, as seen in Fig. 8b, the MLD is much shallower in the simulation with explicit consideration of heat absorption by simulated phytoplankton (INDB) compared to the experiment with constant attenuation coefficient (INDJ). Therefore the

differences with the observational data are substantially reduced in INDB compared to INDJ.

### 3.1.4. Impact of the fully coupled marine biogeochemical variability

*Impact on the water temperature and salinity.* Here we investigate the impact of variable chlorophyll-a concentration when using the corresponding light attenuation parameterization (see Gröger et al., 2013 for details) upon the main oceanic variables by comparing the experiments INDB and INDJ. The vertical distribution of temperature, salinity, dissolved nitrate and

phosphate for different regions of the model domain was already presented in Fig. 6–7 for both experiments. Figure 9 shows the spatial distribution of the difference of the climatological (1975–2004) values of the SST and corresponding standard deviation of the two model runs (INDB-INDJ). In winter (DJF), the use of the light attenuation parameterization based on simulated chlorophyll concentration in INDB leads to a lower SST, which becomes up to 1°C colder in the northern part of the Arabian Sea. The exceptions are the areas near the southwestern coast of India, the northwestern coast of Indonesia and

the eastern part of the Andaman Sea, where an insignificant SST increase which does not exceed 0.1° C can be found. In summer (JJAS) the difference in SST between the two runs is even more pronounced, especially in the northern part of the Arabian Sea and along the eastern coast of India. SST in INDB is also characterized by stronger variability, with a standard deviation of SST approximately 0.3° C higher than in INDJ.





**Figure 9: Spatial distribution of the difference between model runs (INDB-INDJ) for SST (left column) and SST standard deviation (SD, right column). SST and its standard deviation are time-averaged seasonally (DJF, JJAS) for the period 1975–2004.**

When averaging over the annual period (not shown), SST in INDB is also slightly lower and its standard deviation is higher than in INDJ.

Figure 10 shows the spatial distribution of the differences (INDB-INDJ) in DJF and JJAS mean SSS and standard deviation for the same period (1975–2004). Our results show that in all seasonal climatic averages the SSS difference between INDB and INDJ experiments is not strongly pronounced and does not generally exceed 0.2 ‰. The most significant changes in SSS

occur in the Bay of Bengal. Figure 10 also shows that the standard deviation in the two simulations is quite similar, except for the northern part of the Bay of Bengal where the INDB run showed larger seasonal deviations relative to the INDJ experiment.







**Figure 10: Spatial distribution of the difference between model runs (INDB-INDJ) for SSS (left column) and SSS**

**standard deviation (SD, right column). SSS and its standard deviation are time-averaged seasonally (DJF, JJAS) for the period 1975–2004.**

*Impact on the primary production and dissolved nitrate* is shown in Fig. 11 where the differences in depth-integrated modelled phytoplankton primary production (PP) and surface concentration of dissolved nitrate (NO3) are presented. It can be seen that

the PP is higher in the INDB experiment during the main phytoplankton bloom periods (DJF and JJAS). The surface concentration of dissolved nitrates is generally lower in the INDB than in the INDJ experiment. It is especially apparent on the Arabian Sea and agrees well with the increased PP since nutrients are consumed more intensively in the surface layer.



**Figure 11: Spatial distribution of the difference between the model runs (INDB-INDJ) for PP (left column) and NO₃ (right column). PP and NO₃ are time-averaged seasonally (DJF, JJAS) for the period 1975–2004.**

*Impact on water temperature in the ocean upper layers.* To compare the simulated water temperature in the ocean upper layers (up to 100 m depth), we select two complementary regions, where the largest SST difference between INDB and INDJ are found (designated in Fig. 5 as AS: 60-65° E, 20-25° N and BB: 85-90°E, 15-20° N). Figure 12 shows the DJF and JJAS vertical profiles of water temperature (T), shortwave radiation (SW) and phytoplankton concentration (Phyt) for the two experiments averaged over the regions AS, BB and IO. We note a significant cooling of subsurface layers in INDB compared to INDJ.



**Figure 12: Vertical profiles of shortwave radiation (SW), water temperature (T) and phytoplankton concentration (Phyt) in the INDJ and INDB experiments.**

Thermocline dynamics. Thermocline dynamics is among the most important factors mediating the temporal and spatial shape of phytoplankton blooms and their feedback on climate. On the one hand, it acts as a barrier for the vertical exchange between nutrient-depleted surface waters and nutrient-enriched waters from deeper layers and can limit biological productivity. On the other hand, a strong thermocline can effectively reduce the local mixed layer depth and allow phytoplankton to persist longer





within the euphotic layer, thereby increasing the growth rate of marine algae. Moreover, the thermocline has a temperature mediating effect, with a shallower thermocline allowing the surface layer to faster adapt to atmospheric temperatures (e.g., Gröger et al., 2015). The inclusion of phytoplankton into the radiative heat transfer equation alters the vertical distribution of

heat absorption and thus influences the thermocline dynamics. In the following, we compare the thermocline dynamics between the two model runs INDJ and INDB (Fig. 13). A comparison of both runs with WOA 2001 and WOA 2013 data is also discussed here.



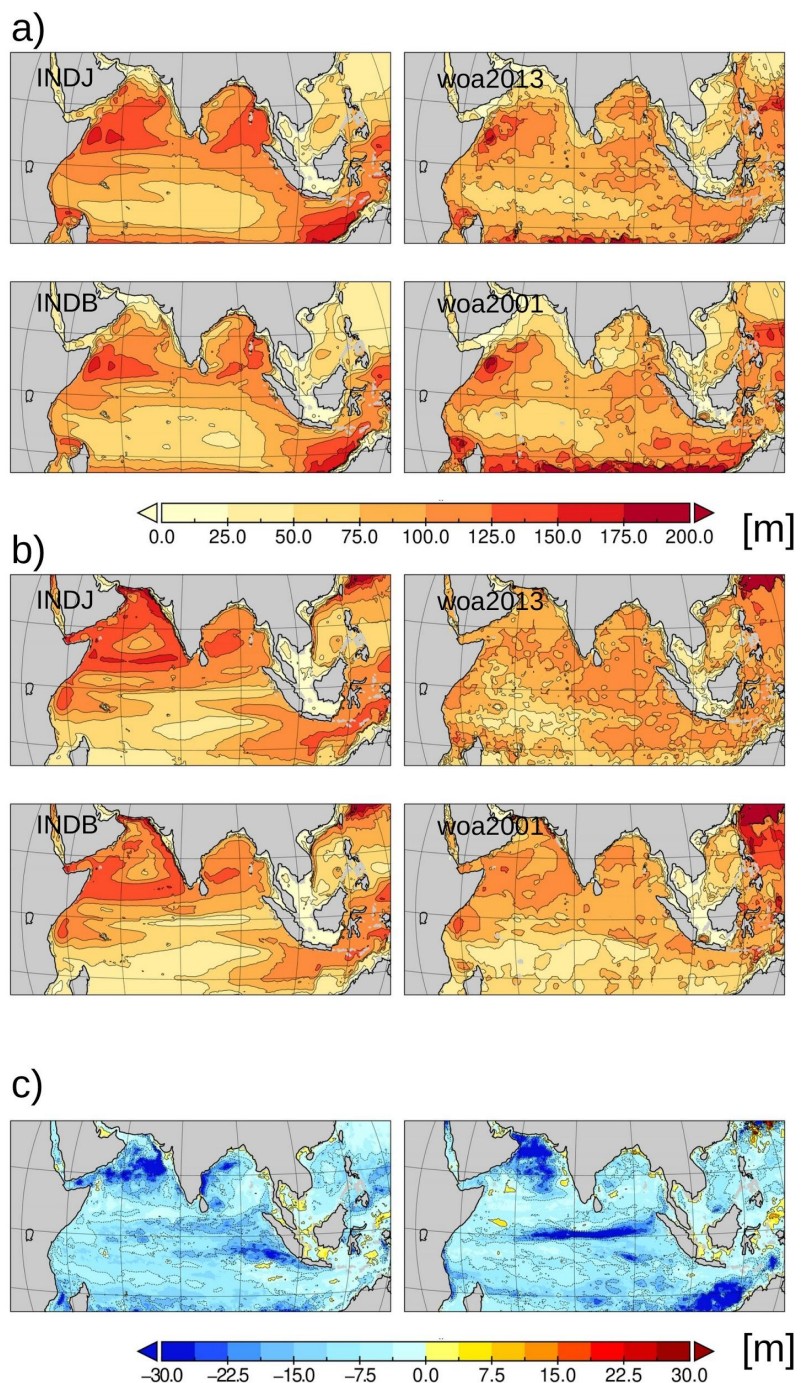

**Figure 13: a) Comparison of simulated summer (JJAS) thermocline depth with thermocline depth derived from WOA data sets. b) same as (a) but for winter thermocline (DJF). c) The difference in thermocline depth between the two runs (INDB-INDJ) for JJAS (left) and DJF (right).**



Data generally tend to be sparse in open ocean regions with less dense measuring campaigns like the Indian Ocean. Then,
caution should be applied when interpreting thermocline dynamics derived from sparse gridded data sets like WOA. Therefore,
we do not provide a quantitative validation here but rather discuss the processes underlying the spatial pattern.

Both simulations and WOA data show distinct gradients in thermocline depth (defined here as maximal temperature gradient
in the water column). During the summer monsoon (Fig. 13a) the thermocline shoals to values smaller than 25 m along the
northern coast of the Arabian Sea and along the Indian coast where moisture carrying SW monsoon winds cause a positive P-
E flux and maintain a vigorous runoff (Ramesh and Krishnan, 2005). Off the Somalian coast and further offshore, the strong
SW monsoonal winds lead to a deepening of the thermocline in wide areas of the open ocean. In both simulations the extension
of this area is larger than in WOA data sets. In the Bay of Bengal, the model simulates a clear east-west gradient with a deeper
thermocline in the east compared to the west. Such a pattern is also observed in WOA to some extent, at least in the WOA
2001. To the south of the equator the thermocline shoals in an extended zonal band with depths well below 50 m. This is
likewise seen in the two WOA data sets though this is less pronounced there. During the winter monsoon, the very shallow
thermocline in the coastal Arabian Sea strongly deepens in response to changed monsoon (Fig. 13b). This seasonal change is
more pronounced in the model simulations but is still significant in the WOA data sets. This indicates that the seasonal
variability is well represented in the model near the coasts.

The simulated thermocline depth is almost everywhere shallower when including the fully coupled biogeochemical variability
in the parameterization of SWR attenuation in the water (Fig. 13c) in both summer and winter. The explicit use of
phytoplankton in the radiated heat transfer (INDB experiment) leads to more heat absorption in the upper layers and less heat
absorption in lower layers. As a result, the thermocline shifts upward compared to the Jerlov type absorption (INDJ experiment)
which follows a simple exponential curve with a constant exponent.

### 3.2 Atmosphere

Here we study the regional distribution of some key atmospheric fields over the South Asia CORDEX region and validate
them for winter (DJF) and summer (JJAS) over the 1975-2004 period. In section 3.2.1 we focus on the regional distribution of
2-meter air temperature (T2M) biases relative to the ERA5 reanalysis (Copernicus Climate Change Service, 2020). Also,
temperature differences between the INDB and INDJ experiments are analysed. This allows us to gain insight into temperature
changes that occur in response to taking into account the variability of ocean biogeochemistry when calculating the SWR
attenuation in the water. In section 3.2.2 the same procedure is followed but taking into consideration the precipitation instead.

### 3.2.1 Air surface temperature

In both seasons, the mean surface temperature in ERA5 is clearly influenced by topography (Figs. 14a, 14d). The lowest values
are reached on highly elevated terrain - especially in winter. The lowest temperatures are attained in world highest mountain
ranges: the Himalaya, Pamir, Hindu Kush and the Tibetan Plateau. The highest summer temperatures are reached along with



the Indo river depression and the Arabian Peninsula. In experiment INDJ the winter daily mean temperatures are simulated
quite well, and biases are relatively small (Fig. 14b and 14e), T2M is underestimated over most of the model domain, except
for its northern and north-western areas where positive biases can reach up to 5 ºC.  The negative biases are mostly below 2
ºC, except for Tibet and Himalaya, where simulated T2M more than 4ºC colder than ERA5 can be found. The largest errors
are found in depressed and/or highly elevated regions and their values may be dependent on factors such as the limited amount

of meteorological stations in topographic highs and lows used for the assimilation in the region and the different representation
of the orography in both REMO and ERA5. JJAS T2M biases are generally lower than in winter, with a similar dependence
on orography. They become positive over most of the Indian subcontinent with maximum values over the northern Indo river
basin where mean temperatures are up to 4 ºC above ERA5. Over the ocean, a positive bias develops in the region where the
monsoon winds are stronger. In general, the most considerable T2M biases are located in regions where larger temperatures

are obtained, pointing to a role of the simulated nocturnal boundary layer and/or radiative fluxes. Like for the SST, ocean
biogeochemical feedback variability leads to a colder surface air temperature over most of the ocean. In DJF the cooling is
stronger over the Arabian Sea and the equatorial strip, reinforcing the weak negative biases already present in INDJ. Over the
land, taking into account the marine biogeochemical variability in INDB slightly improves the cold bias in north-western and
southern India. Still, it leads to a cooling in the central part. The ocean cooling in INDB is also present in JJAS, with stronger

values near the western coast of the Arabian Sea and the Bay of Bengal, downstream of the Monsoon winds.

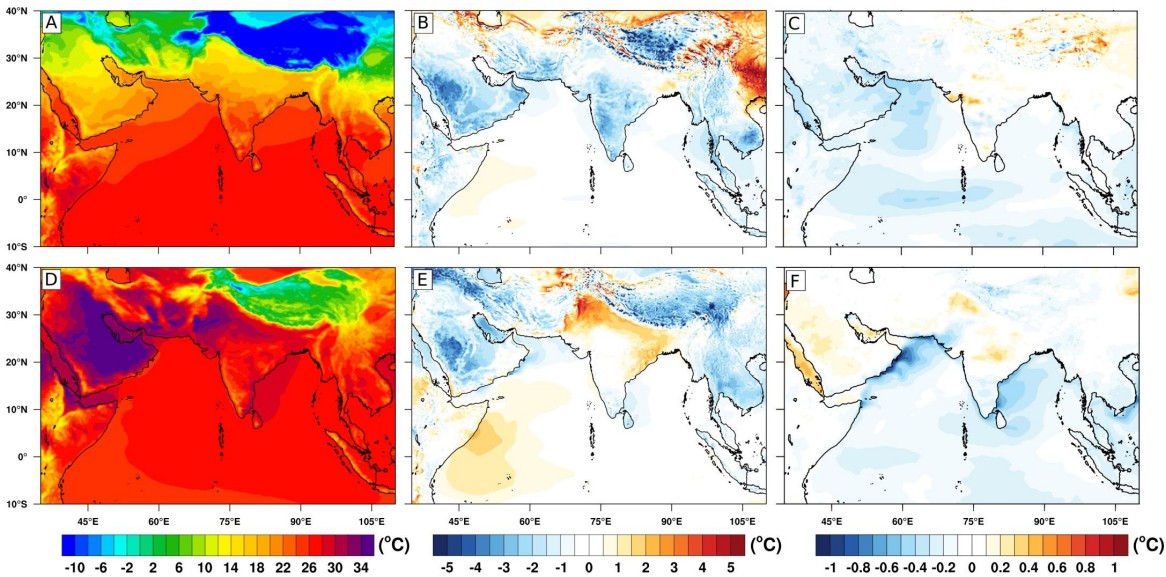

**Figure 14: DJF (upper row) and JJAS (lower row); a) and d) 2-meter temperature ERA5 climatology; b) and e) bias
for the experiment INDJ; c) and f) INDB-INDJ difference.**






### 3.2.2 Monsoon precipitation

The monsoon season in Southern Asia is shaped by different processes which change the atmospheric circulation due to the summer strengthening of the ocean-land temperature contrast. The spatial structure of the observed monsoon precipitation is characterized by two regions of strong rainfall over land (Fig. 15a). The first is located windward, over the western coast of

India, Myanmar and the southern side of the Himalayas. The second region which covers Bangladesh, central India and the eastern coast of the Indian peninsula is the area of maximum monsoon precipitation over the land. The precipitation is weaker over the northwest of India and Pakistan (Kumar et al., 2013). The intricate orography and physical mechanisms involved make the simulation of the monsoon precipitation a difficult task both for global and regional models   (Lucas-Picher et al., 2011). However, stand-alone simulations with REMO have shown to be able to reproduce spatial monsoon precipitation

patterns rather well, although a better quantitative agreement is desirable (Kumar et al., 2014). For instance, the precipitation over the south and central India is overestimated, while the precipitation over the Indo-Ganges plain is strongly underestimated. A wet bias is usually found over the Bay of Bengal and the southern Indian Ocean.

As shown in previous versions of the model (Kumar et al., 2014, Paxian et al., 2016), ROM is able to improve the performance of REMO, simulating more realistic precipitation. The coupling reduces the magnitude of the biases, especially in the regions

where REMO has the most substantial biases, near the eastern coasts of the Arabian Sea and the Bay of Bengal (Fig. 15b). It should be noted that in both INDJ and INDB experiments ROM is forced by MPI-ESM and the biases of the driving ESM influence the results (e.g., Cabos et al., 2020).

Besides the total precipitation, in Fig. 15d-e we show the convective (thereafter APRC) and in 15g-h the large scale (thereafter APRL) component of the precipitation. We can see that in INDJ the main contribution to the biases over the ocean near the

eastern coast of the Arabian Sea comes from APRC, while over the coastal land the main contributor is APRL. The opposite is true for the eastern coast of the Bay of Bengal, especially in Myanmar where the main contributor over the ocean and the coastal regions is APRL, with a lesser contribution from APRC. To the south of the equator, between $10^{\circ}$ S and the equator, both components give a contribution of similar magnitude, albeit the large scale is stronger. Here, both components show a similar displacement of the region of maximum precipitation to the south. While the magnitude of the convective precipitation

is lower than in ERA5, the large scale component is stronger and more zonal than the ERA5 large scale precipitation.



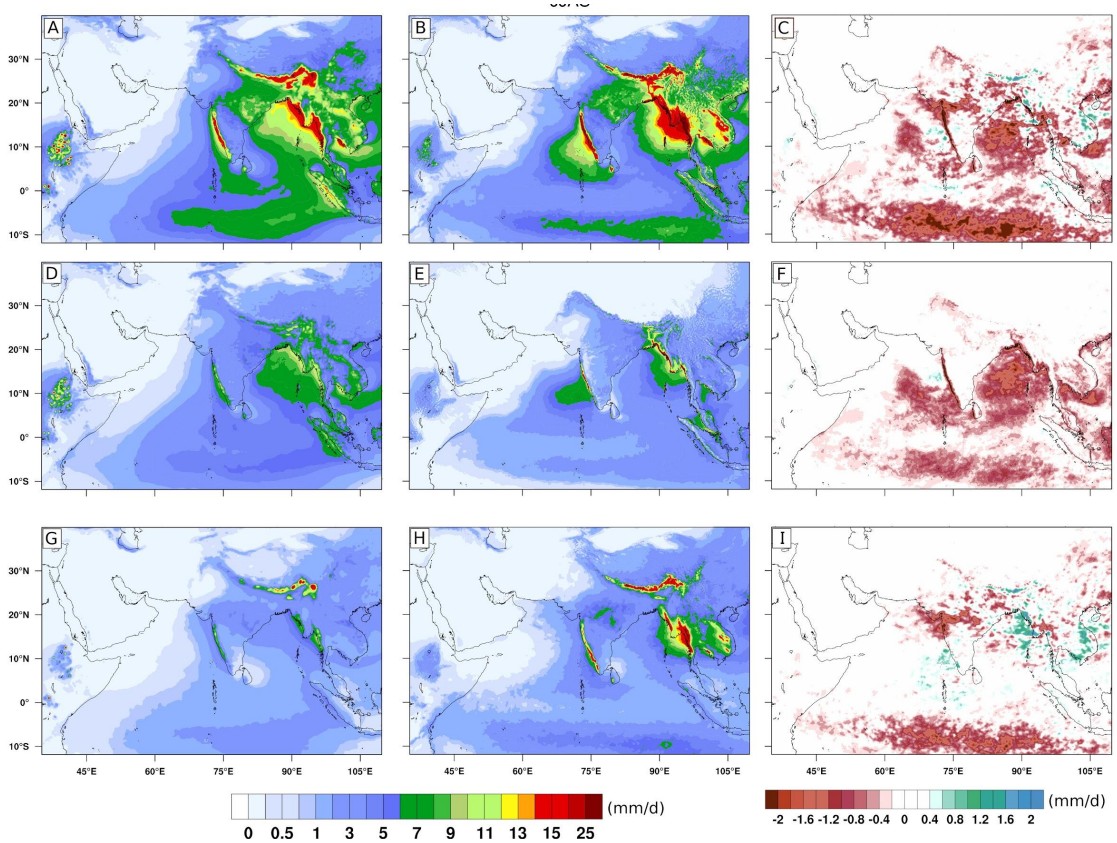

**Figure 15: JJAS precipitation for ERA5 (left column), INDJ experiment (middle column), and the differences between INDB and INDJ (right column) for total precipitation (upper panels); convective precipitation (middle panels), and**

**large scale precipitation (lower panels).**

In the INDB experiment, taking into account the variability of marine biogeochemistry when calculating SWR penetration into the water leads to drying over most of the ocean, especially over the Bay of Bengal, central and north-eastern the Arabian Sea and the strip south of the equator. An apparent reduction in precipitation can also be found inland, along the western Indian

coast. As seen in Fig. 15F and 15I, the contribution of convective and large scale components to these differences vary along the regions.

In the INDB experiment, APRC gives the main contribution (in terms of precipitation) over the Bay of Bengal, the central part of the Arabian Sea and the coastal regions of western India. APRL gives the main contribution to the drying in northern-central India, while in Myanmar it causes a wetting, thus offsetting the impact on APRC. To the south of the equator, between 10° S

and 0°, the impact is similar to both precipitation components.



## 4 Discussion

The effect of the spatial and temporal variability of a fully coupled marine ecosystem upon SWR attenuation in water (experiment INDB) leads to a cooling of the ocean waters compared to a reference INDJ experiment where a constant attenuation coefficient was set equal to 0.06 m⁻¹ (Jerlov IB water type). Generally, taking into account the phytoplankton when calculating light extinction in the ocean leads to a warming of the upper ocean layer and cooling of sub-surface layers compared to a 'no-bio' reference experiment (e.g., Nakamoto et al., 2000; Lengaigne et al., 2007; Park et al., 2014a; Park & Kug, 2012; Park et al., 2014). But, as emphasized in (Lengaigne et al., 2007), the sign of the effect is determined by the choice of the reference experiment. If a truly 'no-bio' approach is implemented in a reference experiment (a 'dead ocean' case), then the SST in the experiments where either constant chlorophyll-a concentration or fully coupled biogeochemical model is implemented becomes higher compared to the reference experiment.

Our results show that in INDB the SST and subsurface waters are cooler than in INDJ over most of the model domain. The attenuation coefficient used in INDJ (0.06 m⁻¹) is not equal to a freshwater attenuation coefficient 0.03 m⁻¹ (e.g., Ilyina et al., 2013). In fact, constant in space and time attenuation coefficient 0.06 m⁻¹ used in the INDJ experiment corresponds to oceanic water masses with relatively small amounts of phytoplankton. Still, it will be incorrect to say that SWR absorption in such waters is fully controlled only by fresh water. It means that in the INDJ experiment the ocean absorbs the incoming SWR more actively than if we use the attenuation coefficient equal to 0.03 m⁻¹. This detail is crucial to explain why the water temperature is cooler in the INDB experiment compared to INDJ. The reason for that is the parameterization used for the SWR extinction in INDB. As given in (Gröger et al., 2013), the attenuation coefficient in this parameterization depends on a constant part representing water attenuation (0.03 m⁻¹) and a variable part representing the attenuation by phytoplankton (for details see Appendix A in (Gröger et al., 2013)). Thus, if phytoplankton concentration is low then the final magnitude of the attenuation coefficient in INDB may be smaller than 0.06 m⁻¹. Analysis of the annual-mean and seasonal-mean phytoplankton distribution in the northern part of the Indian Ocean in both experiments and in satellite data (e.g., Fig.3 herein or Fig.2 in (Liu et al., 2013)) has revealed that this is almost always the case - the observed phytoplankton concentrations in the study area are not enough to raise the attenuation coefficient in the INDB experiment up to an exact value of 0.06 m⁻¹ or higher in the domains considered in our analysis (IO, ASF, BBF, etc). Indeed, figure 16 demonstrates the spatial distribution of the calculated attenuation coefficient at the ocean surface averaged annually (ANN) and seasonally (DJF, MAM, JJA, SON) in the INDB experiment, as well as attenuation coefficient obtained by SeaWiFS measurements (SeaWiFS, 2020) and averaged in the same way.



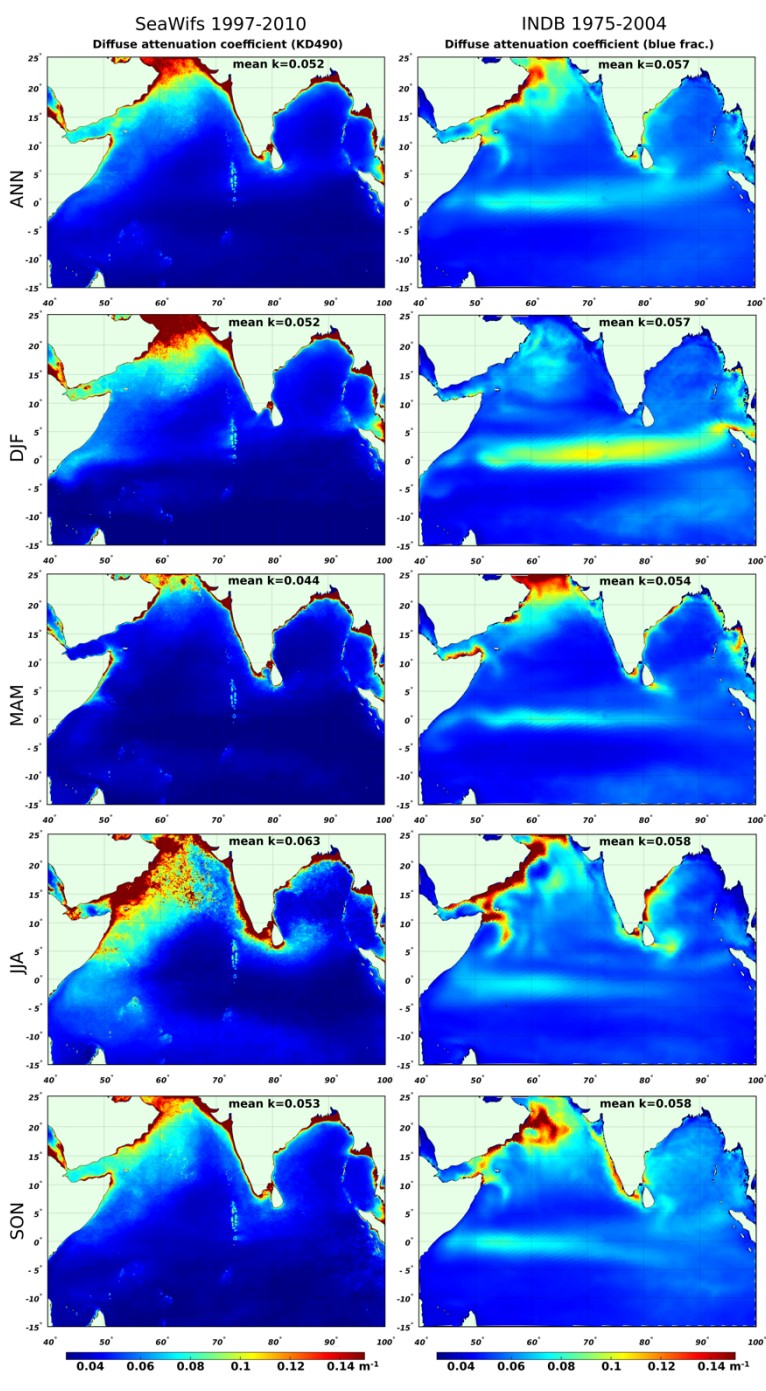

**Figure 16: Spatial distribution of the attenuation coefficient at the ocean surface averaged annually (ANN) and seasonally (DJF, MAM, JJA, SON). Left column - SeaWiFS data averaged over 1997-2010; Right column - INDB results averaged over 1975-2004. For all figures, a mean value of attenuation coefficient (inside the domain) is presented.**






The climatic annual-mean values of attenuation coefficient in the INDB experiment (0.057 m⁻¹) and in SeaWiFS data (0.052 m⁻¹) do not differ too much, taking into account the known uncertainty in determination of this characteristic and not equal periods of averaging (1975-2004 for the model, and only 1997-2010 for SeaWiFS). Compared to the fixed attenuation coefficient used in INDJ, in the INDB experiment, the attenuation coefficient seasonally varies in time and space in a relatively

good accordance with that of SeaWiFS, except for the winter period (DJF). The SeaWiFS's minimum coefficient value occurs in spring (MAM, 0.044 m⁻¹) and maximum occurs in summer (JJA, 0.063 m⁻¹). It corresponds to model results: minimum occurs in spring (MAM, 0.054 m⁻¹) and maximum is in summer and autumn (JJA and SON, 0.058 m⁻¹). Such seasonal variability allows us to examine the influence of marine biogeochemical variability upon the climatic characteristics in the current study. The calculated attenuation coefficient for the blue-green SWR fraction in INDB is smaller than 0.060 m⁻¹ (INDJ),

but the difference is also small: 0.057 m⁻¹ in winter, 0.054 m⁻¹ in spring, 0.058 m⁻¹ in summer, 0.058 m⁻¹ in autumn, with annual-mean being 0.057 m⁻¹.

Figure 12 clearly demonstrates that the water temperature differences in the surface layers between INDB and INDJ experiments are less than the differences between them in the deeper layers. It means that the SWR absorption and vertical water temperature distribution in the INDB experiment follows the same mechanism as reported in other above-mentioned

studies - warming of the surface layer due to additional SWR absorption by phytoplankton and cooling of the subsurface waters, compared to experiment with constant attenuation coefficient. But in our study, we compare INDB results not with a reference experiment with an attenuation coefficient of 0.03 m⁻¹ (without phytoplankton), but with a more realistic experiment with attenuation coefficient equal to 0.06 m⁻¹ (INDJ). The reason for such a choice is that we study not the marine biogeochemical input as it is, but the impact of its variability upon regional climate. That is why for the basic model run in this

study we chose the SWR attenuation scheme with constant attenuation coefficient.

Under the changing climate conditions, the variability of marine biogeochemistry and its corresponding influence on the SWR absorption by the ocean may be an important factor in climate simulations. Thus, the investigation of the differences between the INDJ and INDB experiments presented in this paper is focused on this specific problem. In this respect, making model simulations with fully absent phytoplankton impact on the light absorption while studying regional climate would not make

much sense because such a situation is unrealistic. Using the light attenuation parameterization of the INDB experiment but with Phy=0 would mean to neglect biology completely. However, such additional experiments would help to quantify the effect of phytoplankton in our RESM configuration. Still, taking into account previous studies focused on this matter, significant demands of the presented fully-coupled RESM and limited computational resources, we are compelled to constrain ourselves on the potential research directions in this paper.

Regarding the value of the attenuation coefficient in the reference INDJ experiment equal to 0.06 m⁻¹, such a choice was dictated by the following main reasons. Firstly, previously a lot of studies with the ROM modelling system was made where the reference attenuation coefficient equal to 0.06 m⁻¹ was used (e.g., Sein et al., 2020; Tangang et al., 2020; Zhu et al., 2020; Cabos et al., 2017, 2019; Paeth et al., 2017; Paxian et al., 2016; Paulsen et al., 2018). The choice of this attenuation coefficient





was justified by correctly modeled global primary production, better representation of ITCZ and heat budget, which were in a
good agreement with observations. Since the ocean component of the ROM modeling system is global, we have to use globally-
adjusted model's parameters in the present study as well. Secondly, the choice of the attenuation coefficient in INDJ equal to
0.06 m$^{-1}$ does not assume an unrealistically green ocean. As we have already shown (Fig. 16), the SeaWiFS satellite
measurements give seasonal climatological values of the attenuation coefficient equal to 0.044-0.063 m$^{-1}$ for the considered
domain, with the annual-mean climatic value of 0.052 m$^{-1}$. Finally, (Rochford et al., 2001) reported the assessment of global
attenuation coefficient distribution in the World Ocean based on satellite data. Following their results, the value 0.06 m$^{-1}$ can
be seen as a good estimate of background attenuation coefficient for our domain, except the very coastal waters and Arabian
Sea in august (for details, see Plate 1 and 2 in their paper at the page #30926).

Thus, a correctly adjusted constant light attenuation coefficient can ensure correct global-mean estimates of primary
production, heat budget, etc. Still, it may under- or overestimate SWR attenuation regionally. In this way of thinking, the use
of a completely different parameterization of light attenuation, as is implemented in INDB in this study, is seen as a good
approach for a global model to take into account regional features because it includes spatial- and time varying phytoplankton-
dependent attenuation coefficient.

In the INDB experiment, the thermocline shifts upward compared to the INDJ run where a simple exponential curve of light
attenuation is implemented. This is due to a sharper vertical gradient of water temperature in INDB (Fig. 12) induced by the
non-homogeneity of the vertical distribution of phytoplankton. Hence, in the INDB experiment we see increased light
absorption in the upper ocean layers and decreased - in the subsurface layers, compared to INDJ where a constant attenuation
coefficient controls the SWR absorption.

The different light attenuation parameterization implemented in INDB has cascading effects on model physics, like altered
SST, which further translates into altered atmosphere dynamics. Due to the temporarily varying chlorophyll-a concentrations
in the ocean surface layer and subsequent variable heat absorption, SSTs are by far more variable in the INDB experiment than
in INDJ (Fig. 9).

The higher phytoplankton primary production in INDB (Fig. 11) is most likely the effect of the decreased mixed-layer depth
which allows phytoplankton to prevail longer in the euphotic layer. This effect is more pronounced to the north of 10° N where
the thermocline is relatively deep (and a reduction of the mixed-layer depth in INDB is most effective). In regions where the
thermocline is generally shallower (to the south of 10° N) this effect is of minor importance as light is less limiting there.

During JJAS, the simulated wind in INDJ is slightly weaker than in ERA5 in the Arabian Sea but stronger in the Bay of Bengal
(compare Fig. 17a and 17c).





**Figure 17: JJAS wind (left column) and latent heat (right column) for ERA5 (upper panels); INDJ experiment (middle panels), and (INDB-INDJ) difference (lower panels).**

In the latter, stronger winds lead to stronger latent heat fluxes, while the opposite is true for the Arabian Sea where the weaker wind is associated with a stronger latent heat (Fig. 17b and Fig. 17d). This points to a different nature of the relationship between wind speed and latent heat in both regions, leading to stronger heat flux in both regions. The monsoon winds bring



drier air into the Arabian Sea because it flows over colder water all the way from the equatorial region. Although the cold bias here leads also to a decrease in surface humidity, as the SST bias is lower, the surface humidity bias is lower. The resulting increase of the sea-air humidity difference overcomes the decrease of the wind, thus giving a stronger latent heat flux. This is not true for the west coast where most of the air comes from land (Wu et al., 2007). In the Bay of Bengal, the increase in latent

heat is mainly associated with the simulated winds which are stronger than in ERA5. In the INDB experiment, the marine biogeochemical variability and corresponding variability of SWR absorption by the ocean causes a further cooling over the basin (Fig. 9) and this cooling causes a further drying over most of the domain, especially over the land in regions that are downstream of the monsoon winds. The drying is related to changes both in convection activity and moisture transport. Figure 18a shows the horizontal transport of cloud water for the INDJ experiment. This figure shows the contribution of the large-

scale circulation to the monsoon rain. The Arabian Sea winds are charged with moisture in their path to the Indian subcontinent and Sri Lanka, contributing to the large scale precipitation in the eastern part of the basin and the coastal regions (Fig. 15h). The wind, which loses moisture over the land, is again recharged in his way over the Bay of Bengal, contributing to the strong precipitation in the eastern part of the Bay of Bengal, Myanmar and southeastern Asia. It is noteworthy the recirculation of cloud water in northeastern India due to the presence of the Himalayan range which influences the amount of precipitation

there. The marine biogeochemical variability and corresponding change of SWR absorption affects the precipitation over the Arabian Sea and the Bay of Bengal in different ways. From one side, it reduces the transport of humidity across the equator towards the eastern part of the basin, reducing the large scale precipitation there and in the adjacent coastal regions, reinforcing the effect of the colder water on the convective precipitation. In the Bay of Bengal it reinforces the transport of humidity, increasing the large scale precipitation, contouring the decrease of convective precipitation due to the SST cooling (Fig 18b).


**Figure 18: JJAS horizontal transport of cloud water in INDJ (a) and INDB-INDJ difference (b)**



## 5 Conclusion

A regional Earth System Model based on the ROM model (Sein et al., 2015) has been implemented for the CORDEX South
Asia region. We use the model to investigate the effect of taking into account the full spatial and temporal variability of the
marine ecosystem while calculating light absorption by water upon the regional climate. Two model simulations are conducted
using CMIP5 historical forcing for the period 1920–2005. They differ only by ocean SWR attenuation parameterizations.

The effect of the spatial and temporal variability of a fully-coupled marine ecosystem upon SWR attenuation in water
(experiment INDB) leads to a water temperature decline in the ocean compared to a reference INDJ experiment where a
constant light attenuation coefficient was set equal to 0.06 m⁻¹ (Jerlov IB water type). Based on the analysis of the annual-
mean and seasonal-mean phytoplankton distribution in the northern part of the Indian Ocean attenuation in both experiments
and in satellite data, we can conclude that the reason for this is the low spatially-averaged phytoplankton concentrations in the
analyzed areas, i.e. concentrations are not enough to raise the attenuation coefficient in INDB (which depends on chlorophyll-
a concentration) up to 0.06 m⁻¹ as used in a reference experiment (INDJ). However, the strength (and direction) of temperature
alteration strongly relates to the Jerlov type chosen for the reference simulation, in agreement with earlier findings (e.g.,
Lengaigne et al., 2007).

Both simulations adequately reproduced the precipitation climatology for all seasons. In particular, the spatial pattern of the
monsoon precipitation is well simulated, albeit with some systematic wet biases which are more assertive over the eastern
parts of the Arabian Sea and the Bay of Bengal and the adjacent coastal regions. We found that the marine biogeochemical
variability in INDB and the corresponding change of SWR absorption also affects the amount of precipitation in the model,
leading to drying over most of the basin in the monsoon season. The associated SST cooling leads in general to a reduction of
the precipitation but affects in different ways the two components of the precipitation. In the Arabian Sea the reduction of the
transport of humidity across the equator leads to a reduction of the large scale precipitation in the eastern part of the basin,
reinforcing reduction of the convective precipitation. In the Bay of Bengal it increases the large scale precipitation, contouring
the decrease of convective precipitation due to the SST cooling.

Thus, in comparison with simulation using a constant light attenuation coefficient (0.06 m⁻¹, Jerlov IB water type), the major
impacts of including the full biogeochemical coupling with corresponding light attenuation in water, which in turn depends on
variable chlorophyll-a concentration, include the enhanced phytoplankton primary production, a shallower thermocline,
decreased SST and water temperature in subsurface layers, with cascading effects upon the model ocean physics which further
translates into altered atmosphere dynamics.

In summary, the presented model demonstrates the locally substantial impact of phytoplankton-related chlorophyll on the
atmospheric climate of the Indian Ocean. However, this study does not take into account the direct impact of biology (i.e.
productivity) on atmospheric pCO2 and the subsequent impact on the atmospheric radiation budget. Because of this, the impact
of marine biology on climate may be underestimated.




**Acknowledgements.**

This work is jointly funded by Russian Science Foundation (RSF), Russia (Project 19-47-02015) and Department of Science and Technology (DST), Govt. of India (grant number DST/INT/RUS/RSF/P-33/G) through a project "Impact of climate change on South Asia extremes: A high-resolution regional Earth System Model assessment". The research was performed in
the framework of the state assignment of the Ministry of Science and Higher Education of Russia (0128-2021-0014). This work used resources of the Deutsches Klimarechenzentrum (DKRZ) granted by its Scientific Steering Committee (WLA) under project ID ba1144. We thank the anonymous reviewers and Andreas Oschlies for the constructive suggestions and critical remarks, which helped to improve the manuscript.

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
