# Peer review of "Indian ocean marine biogeochemical variability and its feedback on simulated South Asia climate"

_Earth System Dynamics, 2021_

## Author Comment (AC1)

**Comment on esd-2021-64**
**Anonymous Referee #1**
Referee comment on "Indian ocean marine biogeochemical variability and its feedback on
simulated South Asia climate" by Dmitry V. Sein et al., Earth Syst. Dynam. Discuss.,
https://doi.org/10.5194/esd-2021-64-RC1, 2021
**Review of Sein et al. entitled "Indian ocean marine biogeochemical variability**
and its feedback on simulated South Asia climate"**

**General comments:**
Based on two numerical experiments run with the ROM model, Sein et al. investigate how the attenuation of the incoming radiation by chlorophyll may perturb the ocean response.
Then they evaluate the resulting feedback from the altered ocean on the coupled ocean atmosphere system. This topic is of interest as it sheds light on the uncertainties associated with model configuration when modelling climate and ocean-atmosphere interactions.

*We thank the reviewer for his kind feedback and valuable comments regarding the figures and text, which allowed us to improve the presentation of our results. New versions of figures have been inserted into the manuscript*

**Major comments:**
The scientific questions are of great interest, and the paper is in general well-structured and well-written.

**Figures:**
I recommend to put more care on the readability of your figures. Below are few examples of the improvements you may bring to them:
Figure 2: to add a title per row on the plot (DJF or JJAS) would help the reader to quickly understand the results.

*We have added these titles to the plot.*

Figure 8 has no x- and y-ticks.

*We have added longitude and latitude values on the maps shown in Figure 8.*

Figure 13 has no x- or y-ticks. Maybe a) JJAS, b) DJF, and c) (INDB-INDJ) with inside subtitle JJAS (left) and DJF (right).

*Figure 13 has been revised and fragments containing WOA 2001 have been removed. The relevant text (marked in yellow) has been edited. Below new Figure 13 is shown*

[Figure]

*Figure 13: a) Comparison of simulated thermocline depth with thermocline depth derived from WOA 2013 data sets for JJAS (left) and DJF (right). b) The difference in thermocline depth between the two runs (INDB-INDJ) for JJAS (left) and DJF (right).*

Figures 14, 15 and 17: This is a cosmetic suggestion that you may take or not into account: I think it would improve the readability of those Figures to put a title for each column, and each line. Something like (this example is done for Figure 15):
ERA5 INDJ (INDB-INDJ)

Total A B C
convective D E F
large-scale G H I

*We have rearranged Figures 14, 15 and 17 in accordance with the recommendations of the reviewer, and, accordingly, the captions under the figures have been changed.*

In general, for most of your Figures, ticks are too small.

*We increased ticks on most of the Figures, but for some we did not change them. Based on our experience, this issue is handled by a technical editor. For the sake of a curiosity: we recently encountered a situation when, after increasing ticks at the request of a reviewer, we had to change them to the old ones, i.e. reduce, at the request of a technical editor.*

**Minor comments:**
l. 114: "The oceanic component of ROM is the Max Planck Institute Ocean Model" -> On the basis of what I understand I would suggest to write: "The oceanic component of ROM is the global Max Planck Institute Ocean Model". When reading your model description it is not straightforward to understand that your oceanic model has a global configuration, and that's why "MPIOM provides the possibility to refine the grid resolution in the region of interest and to avoid the lateral boundary conditions in the ocean while performing
calculations". Please clarify.

*We have changed the statement in accordance with the comment of the reviewer:*
*The oceanic component of ROM is the global Max Planck Institute Ocean Model.*

l. 125: Please specify how many vertical levels has your ocean configuration.

*We have added the required information to the manuscript:*

*MPIOM has 40 vertical z-coordinate levels with the following thicknesses:16, 10, 10, 10, 10, 10, 13, 15, 20, 25, 30, 35, 40, 45, 50, 55, 60, 70, 80, 90, 100, 110, 120, 130, 140, 150, 170, 180, 190, 200, 220, 250, 270, 300, 350, 400, 450, 500, 500, 600.*

l. 166: WOA13 ? why don't you use the latest release WOA2018 ?

*We use WOA13 because:*
*1. WOA13 is widely used in the scientific literature and thus can be easily compared to other studies that use the same reference.*
*2. As stated in the WOA18 description (https://www.ncei.noaa.gov/data/oceans/woa/WOA18/DOC/woa18documentation.pdf) "WOA18 temperature and salinity data is still published as preliminary in order to take advantage of community-wide quality assurance and comments". Therefore, we would avoid confusion with future studies using WOA18 final version.*
*We have added this explanation into the text together with a brief description of WOA13 and other datasets after L.155:*

*We compare the calculation results for both experiments (INDJ and INDB) with the best observational datasets to date for the region under consideration: they include oceanographic data compiled in the World Ocean Atlas 2013 (WOA13, Levitus et al., 2014) and the satellite data from the Sea-viewing Wide Field-of-view Sensor (SeaWiFS) and Moderate-resolution Imaging Spectroradiometer (MODIS) Terra. The WOA13 is a set of climatological mean, gridded fields of oceanographic variables based on in-situ measurements from a wide variety of sources. Global, decadal averages of temperature, salinity, oxygen and nutrients are provided at monthly, seasonal and annual averaging periods on 102 standard depth levels from 0 to 5500m, and at 0.25 ° (temperature, salinity) and 1 ° (all variables) horizontal resolutions. We do not use the latest edition of WOA18 (Boyer et al., 2018) for the following reasons: 1) WOA13 is widely used in the scientific literature and thus can be easily compared to other studies that use the same reference, 2) as stated in the WOA18 description (https://www.ncei.noaa.gov/data/oceans/woa/WOA18/DOC/woa18documentation.pdf) WOA18 temperature and salinity data is still published as preliminary in order to take advantage of community-wide quality assurance and comments.*

l. 169: "NE winds": north-easterly winds ? Define "NE".
*NE is defined.*

l. 171: "SW winds": south-westerlies ? Define "SW".
*SW is defined.*

l. 171: I find a bit strange to use 3 months for winter mean (DJF), but 4 months for the mean of the summer season (JJAS). Please explain briefly why, or change it.

*According to long-term observations of the Indian Meteorological Department, the winter season with prevailing north-easterly winds in South Asia lasts three months (December, January and February), while the rainy monsoon season, during which the south-westerly wind prevails, lasts four months from June to September (see, for example, Kumar et al., 2013). That is why we, like almost all other researchers of the weather and climate of South and Southeast Asia and the northern Indian Ocean, use the averaging of the winter season over three months (DJF), and the summer season over four months (JJAS)*

*Reference:*
*Kumar P, Wiltshire A, Mathison C, Asharaf S, Ahrens B, Lucas-Picher P, Christensen JH, Gobiet A, Saeed F, Hagemann S, Jacob D. Downscaled climate change projections with uncertainty assessment over India using a high resolution multi-model approach. Sci Total Environ. 2013 Dec 1;468-469 Suppl:S18-30. doi: 10.1016/j.scitotenv.2013.01.051. Epub 2013 Mar 28. PMID: 23541400.*
   *To explain more clearly why we are using such durations of seasons in the region in question, the text at the beginning of paragraph 3.1 will be edited as follows:*

*According to long-term observations of the Indian Meteorological Department, we distinguish the following seasonal periods used for the verification procedure based on the monsoon activity in South Asia and in the northern part of the Indian Ocean:*

l. 190: Following your draft structure, you have to put in italics "Sea surface concentration of dissolved nitrate".

*Corrected.*

l. 250: given that the amplitude of your Chl-a improves when using the variable C:Chl ratio, are you going to try to replace the fixed C:Chl ratio by the parameterization of Anderson directly in HAMOCC in a future study?

*Yes, we plan to do it in the near future*

l. 219: "While these changes are in accordance with the changing wind regime, the satellite data also shows higher concentrations during summer." In its actual form the sentence is not easy to interpret for me. I would mark more clearly the opposition between your model results that are in accordance with the wind regime but not with the satellite data. -> something like " These modeled Chl-a changes are in accordance with seasonal changes of the wind regime. However the satellite data show high concentrations during summer in that coastal area that our model did not represent."

*We have changed the text in accordance with the recommendation of the reviewer as follows:*
*In winter the model simulates enhanced chlorophyll-a concentrations along the eastern boundary of the Bay of Bengal while showing their decrease during monsoon season. These modeled chlorophyll-a changes are in accordance with seasonal changes of the wind regime. However, the satellite data show high concentrations during monsoon season in that coastal area that our model did not represent.*

l. 220: "The most plausible explanation for this is a persistently high supply of riverine nutrients around the year." Do you want to say that this persistent high supply is not represented in your model ? From l. 139-141 I understand that you did not represent riverine inputs: "Secondly, coastal characteristics, especially in front of large rivers with high nutrient load and limited exchange with the open ocean, are not resolved which is however crucial in high resolution downscaling simulations." Please, clarify.

*The Reviewer is correct, our model does not represent the riverine input of nutrients in the current study, as stated in L139-141 of the manuscript. In L220 we tried to say that the persistent high supply of nutrients (which exists in reality and is not represented in our model) may be responsible for increased surface chlorophyll-a concentration visible from the satellite images (Fig. 3). To eliminate the ambiguity and to clarify the text, we have rephrased L220 as follows:*
*The most plausible explanation for this is a persistently high supply of riverine nutrients around the year which occurs in reality and which is not specified in our model.*

Figure 3: Surface Chl-a in DJF show a strong equatorial tongue not presents in the observations. In l. 223-224 you stated that "the enhanced model's equatorial surface phytoplankton concentration cannot simply be related to incorrect wind simulation". However I wonder if it may not be inherited from the forcing fields outside the modeled domain? If I have correctly understood your coupled configuration, the atmospheric forcings outside your simulated region come from MPI-ESM, and so may imprint direct biases to 1) the atmospheric fields inside your simulated domain (but you found "good agreement, both qualitative and quantitative, between model's winds and ERA5's winds"), as well as 2) to the ocean outside the modeled domain. Then 1) and 2) would indirectly affect the ocean inside the modeled domain.

*The atmospheric forcing outside the coupling area is taken from a CMIP5 20th century simulation with the MPI-ESM LR. The modeled atmospheric fields in our study, specifically, wind velocity, is in a good accordance with ERA5 data (comparison is presented in Fig. 17). Thus, the modeled wind field inside the coupled area does not have significant errors inherited from the wind field outside the coupled area. We believe that the MPI-ESM atmospheric fields which drive the global ocean in our configuration are correct. What is left - is the ocean model itself, which may simulate the equatorial ocean dynamics not quite realistically. The reason of that, in our opinion, is stated in the manuscript in L224-226:*
*The problem may be related to the relatively coarse vertical resolution of MPIOM in the upper layer (16 m) together with a simple turbulence closure scheme in MPIOM based on (Pacanowski and Philander, 1981).*

Regarding forcings description:
l. 127: "The model is driven by data from a CMIP5 20th century simulation with the MPIESM LR setup." Following my previous comment, I would have rather written: "The model is driven by atmospheric data from a CMIP5 20th century simulation with the MPI-ESM LR setup."
*We have changed the text in accordance with the recommendation of the reviewer*

l 450-451: "It should be noted that in both INDJ and INDB experiments ROM is forced by MPI-ESM and the biases of the driving ESM influence the results (e.g., Cabos et al., 2020)." Here also I would specify "[…] and the atmospheric biases of the driving ESM […]".
*We have changed the statement according to the referee's suggestion. Now it reads as follows:*

*It should be noted that in both INDJ and INDB experiments ROM is forced by MPI-ESM and the **atmospheric** biases of the driving ESM influence the results (e.g., Cabos et al., 2020).*

l. 224: "The problem may be related to relatively coarse vertical resolution of MPIOM in the upper layer (16 m) together with simple turbulence closure scheme in MPIOM based on (Pacanowski and Philander, 1981)." For me Pacanowski and Philander (1981) allow to diagnose vertical mixing coefficients from the large scale variables computed by the model. Does it mean that you are not really taking advantage of your configuration at 15km of horizontal resolution (by using parameterizations not suitable for your resolution) ?

*Partially it is true and we are going to replace the PP mixing scheme. On the other hand, the model has 15 km resolution near the poles only. In the part of the Indian Ocean included in the South Asia CORDEX domain the resolution ranges from 23.3 to 24.5 km.*
*We have inserted an explanatory text (L.125-126) as follows:*

*In this work, we use for REMO the slightly enlarged South Asia CORDEX domain (http://www.cordex.org), while for MPIOM the global mesh has a variable horizontal resolution which reaches up to 15 km inside the coupled region and ranges from 23.3 to 24.5 km in the part of the Indian Ocean included in this domain (Fig. 1).*

l. 225: "The overestimation or underestimation of ocean productivity along the equatorial divergence zone is a common problem of many ocean general circulation models (e.g.,

Steinacher et al., 2010)." By reading the paper of Steinacher I rather get the feeling that only 2 models had this problem (MPIM and CSM1.4), in which is yours.

*The reviewer is right: we have too generalized the models that face this problem. This phrase is now given in the text of the manuscript as follows:*

*The overestimation or underestimation of ocean productivity along the equatorial divergence zone is a problem also in one of the best models of the earth's climate system, CSM1.4 (Steinacher et al., 2010).*

l. 257: "However, during several short periods the MODIS's daily-mean climatic concentrations (MODIS, 2020) appear to be higher than in the model." Are model time series based on daily or monthly outputs ? I guess they are daily, because if based on monthly outputs it would be not surprising that MODIS daily-mean concentrations exceed model outputs. Could you please clarify the modelled outputs frequency you use for that analysis?

*The blue and red curves (Chla INDJ and Chla INDB) are daily-mean climatic averaging of modeled surface chlorophyll-a concentration for the period 1997-2005.*
*This clarification has been added to the caption of Fig. 4.*

Figure 4: I have some reservations superimposing MODIS and SeaWifs observations on that figure. I understand that you want to validate your model configuration with integrity and also show that observed daily concentrations may be higher than your model prediction. However, I think that these observational seasonal cycles would be valuable to show only if in good accordance with your modeled seasonal cycle, which is not particularly the case (except maybe in the Somali upwelling area). I am afraid it discredits your configuration, while the sensibility experiment you performed and associated effects are of great interest. I guess that you tried to compare your model and observations on a more aggregated region, and that was not better ?

*For the Fig. 4's comparisons, we used 1x1 degree lon-lat squares that cover the whole domain. For each of these small areas, we plotted the time-series of several model fields (e.g., MLD, SST, Chla concentration, etc). We did not analyze modeled mean surface chlorophyll-a concentration averaged for the entire model domain (or areas larger than 1x1 degree) because chlorophyll-a concentration is very spatially-variable. Thus, the spatial-averaging over a large area would give a very smoothed pattern, without peak concentrations, etc. The locations in the Arabian Sea, Somali upwelling area, and the Bay of Bengal shown in Fig. 4, were chosen as one the most interesting areas in terms of ocean primary production in the northern Indian Ocean. Unfortunately, the quality of comparisons in other areas (squares 1x1 degree) is approximately the same.*
*We fully agree with the Reviewer that the time-series of modeled surface chlorophyll-a concentration presented in Fig. 4 are not in a good accordance with those obtained from satellite measurements. As emphasized in the manuscript (L232), the main cause of such discrepancy between the model and satellite data may be the relatively simplicity of the marine biogeochemical model HAMOCC:*

*The overestimation of chlorophyll-a concentration in the domain may also be explained by a relatively simple description of phytoplankton dynamics in the HAMOCC model. HAMOCC includes only one type of phytoplankton and, as a component of a global climatic model, it was*

*configured to produce realistic global-mean primary production (Ilyina et al., 2013), but may significantly over- or underestimate some regional features of marine biological productivity. We suppose that this is the main cause of differences between satellite and model results.*

*Despite the poor correspondence between the model and satellite data presented in Fig. 4, when preparing the manuscript, we decided to keep this results and include them in the final version in order to demonstrate the model's biases which may be responsible for uncertainties in SWR attenuation in the water by phytoplankton. This result also demonstrates that a globally-tuned marine biogeochemical model which adequately simulates the global ocean primary production, may give large biases on a smaller regional scale.*

l. 261: "time series" -> I suggest to write rather "seasonal cycle".

*We agree with the Reviewer in that, strictly speaking, the curves and points plotted at Fig. 4 are not time-series, but plots of specifically-processed data aimed at showing the average (climatic) changes within the annual cycle at some characteristic locations of the study area. Therefore, we have excluded the term "time series" from the text and have modified the Fig. 4 caption accordingly:*

*Figure 4: Comparison of the simulated (INDJ and INDB) and observed (SeaWiFS, MODIS Terra) surface chlorophyll-a concentration in the Arabian Sea (a), Somali upwelling area (b), and the Bay of Bengal (c). DC and MC - daily-mean and monthly-mean climatic averaging of satellite data for the period 1997-2005, respectively. The blue and red curves (Chla INDJ and Chla INDB) are daily-mean climatic averaging of modeled surface chlorophyll-a concentration for the period 1997-2005.*

Figure 6: I am surprised that your comparison with observations for temperature and salinity are not better with a regional configuration. By curiosity, have you tried to force an INDB-type configuration by a reanalyse product?

*No, we did not try to run uncoupled INDB-type configuration forced by a reanalysis. The point is that sea surface temperature in uncoupled mode strongly depends on prescribed 2 meter air temperature and in this case it is quite difficult to obtain biogeochemical effects.*

l. 367 put in italics "Thermocline dynamics".
    *Corrected.*

l. 376: WOA 2013 vs WOA 2001 -> why don't you compare data of WOA2013 with the latest release WOA2018 (https://www.ncei.noaa.gov/products/world-ocean-atlas) rather than with an older one (WOA2001) ?
Boyer, Tim P.; Garcia, Hernan E.; Locarnini, Ricardo A.; Zweng, Melissa M.; Mishonov, Alexey V.; Reagan, James R.; Weathers, Katharine A.; Baranova, Olga K.; Seidov, Dan; Smolyar, Igor V. (2018). World Ocean Atlas 2018. NOAA National Centers for Environmental Information. Dataset. https://accession.nodc.noaa.gov/NCEI-WOA18.

*We explained above (l. 166) why we don't use WOA18. We also removed the old release WOA2001 from the Figure 13 and the relevant text because we did not make a detailed comparison of this release with the WOA2013.*

l. 402-403: Fig 13c for DJF suggests that the equatorial overestimation of CHLa in INDJ (Figure 3) may be due to a too deep thermocline depth when considering a constant

light attenuation coefficient. But you did not present or discuss surface CHLa differences between experiments (INDB-INDJ), why ? PP differences (Figure 11) show a small increased production in the equatorial area in DJF, but it seems weak. On the contrary your Figure 16 suggests that the equatorial bias is still present in your INDB simulation. Please could you discuss that aspect ?

*The following figure demonstrates the difference (INDB-INDJ) for modeled surface chlorophyll-a concentration for the DJF season. It shows that the positive bias in surface chlorophyll-a concentration is also present in the INDB experiment along the equator, as well as in Fig. 11 in the manuscript (primary production). This was the reason why we did not include this figure in the manuscript - the difference in primary production distribution (INDB-INDJ) shown in Fig. 11 is rather descriptive. Summarizing, both modeled primary production and surface chlorophyll-a concentration, in both experiments INDB and INDJ, demonstrate that the model's description of equatorial processes related to ecosystem functionality in this band is still rather poor, as discussed in the manuscript (L224-244).*

[Figure]

Chl-a INDB-INDJ (1975-2004), season: DJF

*Consequently, the equatorial bias for water attenuation coefficient visible in Fig. 16 (a new one!, see below) for DJF season, is due to the erroneous simulation of phytoplankton concentration in the equatorial band. As follows from the above-shown picture and Fig. 3 in the manuscript, this erroneous phytoplankton field along the equator occurs in both INDJ and INDB experiments. In our opinion, it is related to the shortcomings of the biogeochemical model HAMOCC, and to the coarse vertical resolution of MPIOM in the upper ocean layer which, together with Pacanowski & Philander (1981) turbulent mixing parameterization, do not give realistic results of equatorial dynamics simulation and, hence, phytoplankton surface concentration that strongly depends on nutrient supply.*

[Figure]

***Figure 16: Spatial distribution of the attenuation coefficient at the ocean surface averaged annually (ANN) and seasonally (DJF, MAM, JJAS, ON). Left column - SeaWiFS data (NASA Goddard Space Flight Center ..., 2021c) averaged over 1997-2005; right column - INDB results averaged over 1997-2005. For all figures, a mean value of attenuation coefficient (inside the domain) is presented.***

l. 412: "In both seasons, the mean surface temperature in ERA5 is clearly influenced by topography (Figs. 14a, 14d)." On that topic I suggest you to read and cite Samson et al. (2016). They show that land surface temperature errors are a major source of low-level circulation and rainfall biases for your modelled region. Your Figure 14E shows the bias they describe for JJAS with a cold bias over the Middle-East (impacting the Findlater jet) and a warm bias over India (although yours appears restricted to the north of India: see their Figure 2c).
Samson, G., Masson, S., Durand, F. et al. Roles of land surface albedo and horizontal resolution on the Indian summer monsoon biases in a coupled ocean–atmosphere tropical-channel model. Clim Dyn 48, 1571–1594 (2017).
https://doi.org/10.1007/s00382-016-3161-0

*Thanks for the reference, it indeed gives a good insight into the mechanisms through which resolution influences the biases and, most importantly, to the role played by the surface albedo in the generation of the biases. The referred is changed as follows:*

*In both seasons, the mean surface temperature in ERA5 is clearly influenced by topography (Figs. 14a, 14d). In JJAS the cold bias over the Middle-East and the warm bias over India (Fig.14e) impact the strength and the path of the Findlater jet (Samson et al, 2017).*

*We have also added the reference on Samson et al. to the references list.*

l. 424-425: "In general, the most considerable T2M biases are located in regions where larger temperatures are obtained, pointing to a role of the simulated nocturnal boundary layer and/or radiative fluxes". I suggest to add the role of the good representation of the land surface albedo (impacting surface heat budget, winds and precipitations) described in Samson et al. (2016).

*We thank, again, the referee for this very useful reference. Actually, we did a tuning in our land surface model in order to reduce this effect. We have made the suggested improvements on the manuscript. The statement now reads as follows:*
*"In general, the most considerable T2M biases are located in regions where larger temperatures are obtained, pointing to a role of the simulated nocturnal boundary layer and/or radiative fluxes. As shown in Samson et al. (2017), radiative fluxes (as well as winds and precipitation) are influenced by the representation of the land surface albedo and this influence is important in this region".*

l. 453: "and in 15g-h" -> and in Fig. 15g-h
  *Corrected.*

l. 511-515: You wrote "The climatic annual-mean values of attenuation coefficient in the INDB experiment (0.057 m-1) and in SeaWiFS data (0.052 m-1) do not differ too much" and "Compared to the fixed attenuation coefficient used in INDJ, in the INDB experiment, the attenuation coefficient seasonally varies in time and space in a relatively good accordance with that of SeaWiFS, except for the winter period (DJF)." However amplitude along the Indian west coast and the coast surrounding the Bay of Bengal are not so comparable. And spatial patterns at the equator and along the Somalia upwelling are quite different. Please rephrase.

*We have replotted Figure 16 and following the Reviewer's suggestion, have changed this part as follows:*

*The climatic annual-mean value of attenuation coefficient in the INDB experiment (0.056 m⁻¹) is closer to that in SeaWiFS data (0.051 m⁻¹) than the INDJ's value (0.06 m⁻¹). Compared to the fixed attenuation coefficient used in the INDJ experiment, in the INDB experiment the attenuation coefficient has its temporal and spatial variability which roughly replicate that of SeaWiFS, except for the winter period (DJF), taking into account the known uncertainty in determination of this characteristic. The SeaWiFS's minimum coefficient value occurs in the pre-monsoon season (MAM, 0.042 m⁻¹) and maximum occurs in the monsoon season (JJAS, 0.059 m⁻¹). It corresponds to model results: minimum occurs in the pre-monsoon season (MAM, 0.053 m⁻¹) and maximum is in monsoon season (JJAS, 0.059 m⁻¹).*

Figures 17 and 18: Please complete your captions to explain that shading shows the wind (cloud water transport) module and arrows show its direction.

*We have changed the figure captions as follows:*

*Figure 17: JJAS wind (left column, arrows are wind velocities, the color scale is the wind speed/module) and latent heat (right column, the color scale is the latent heat flux) for ERA5 (upper panels); INDJ experiment (middle panels), and (INDB-INDJ) difference (lower panels).*

*Figure 18: JJAS horizontal transport of cloud water ( arrows show the vector of cloud water transport, the color scale is its module) in INDJ (a) and INDB-INDJ difference (b).*

l. 523: "Figure 12 clearly demonstrates that the water temperature differences in the surface layers between INDB and INDJ experiments are less than the differences between them in the deeper layers". Vertical profiles in Figure 12 only go down to 100m: have you checked the temperature difference below 100m ? Are you sure it is maximal at 100 m depth? It would be interesting to see a vertical section or profiles up to 500 m depth. I guess the differences extend up to 300m.

*Below are two versions of Figure 12: a) for depths (0, 100m) and b) for depths (0, 300m). As can be seen in Fig. 12b, only in the AS region in the winter season the maximum temperature difference between INDB and INDJ experiments is at a depth of 126 m, in other cases it occurs at depths of 104 m and less. In layers lying below the depth of the maximum temperature difference, it decreases to negligible values at depths of 180-240m. At the same time, the differences between INDB and INDJ in shortwave radiation and phytoplankton concentration, which occur in the layer from 0 to 60m, are distinguishable in Fig. 12a and indistinguishable in Fig. 12b.*
*In order to preserve clarity in the vertical profiles of shortwave radiation and phytoplankton concentration, we left in the manuscript the Figure 12a for depths (0, 100m).*

*A(100m scale, Figure from the manuscript)*

[Figure]

*Figure 12: Vertical profiles of shortwave radiation (SWR), water temperature (T) and phytoplankton concentration (Phyt) in the INDJ and INDB experiments.*

B (300m scale)

[Figure]

**Vertical profiles of shortwave radiation (SW), water temperature (T) and phytoplankton concentration (Phyt) in the INDJ and INDB experiments. Maximum temperature difference between INDB and INDJ experiments (dTmax) and its depth are indicated on each fragment above the legend**

*At the same time, answering the reviewer's question about the behavior of water temperature at depths exceeding 100m, we changed the description of the vertical profiles of the water temperature in INDB and INDJ experiments as follows (l. 523):*

*Figure 12 clearly demonstrates that for the upper 100-meter layer the water temperature differences between INDB and INDJ experiments in the surface layers are less than the differences between them in the deeper layers. Consideration of further changes in this difference with increasing depth showed that the maximum difference occurs at depths of about 100m or less and in layers lying below the depth of the maximum difference, it decreases to negligible values at depths of 180-240m.*

l. 536: "Phy=0" ? What does it mean ? With fully absent phytoplankton I guess. Please, clarify the text.

*We have changed this text into:*

*Using the light attenuation parameterization of the INDB experiment but with fully absent phytoplankton, would mean to neglect biology completely.*

---

## Author Comment (AC2)

**Comment on esd-2021-64**
**Anonymous Referee #2**
Referee comment on "Indian ocean marine biogeochemical variability and its feedback on
simulated South Asia climate" by Dmitry V. Sein et al., Earth Syst. Dynam. Discuss.,
https://doi.org/10.5194/esd-2021-64-RC2, 2021

**General Comments:**

The paper presents an interesting study on marine biogeochemical variability in the Indian Ocean and its impact on regional climate. The authors investigate the effect of a static and a variable marine biogeochemical light absorption on simulating the regional climate on the Indian Ocean. Both atmosphere and ocean present climate are evaluated against observations, as well as the quality of physical and biogeochemical characteristics simulated. The paper is well written and organised.

*We are grateful to the reviewer for his kind feedback and valuable comments, which allowed us to improve the text of the manuscript. Changes made to the manuscript are highlighted in yellow. They are shown in italics in responses to comments.*

**Specific Comments:**

Methods:
Please include the horizontal and vertical resolutions of your simulations.

*A description of horizontal and vertical resolutions is included instead of L.125-127 as follows:*

*In this work, we use for REMO the slightly enlarged South Asia CORDEX domain (http://www.cordex.org), while for MPIOM the global mesh has a variable horizontal resolution which reaches up to 15 km inside the coupled region and ranges from 23.3 to 24.5 km in the part of the Indian Ocean included in this domain (Fig. 1). MPIOM has 40 vertical z-coordinate levels with the following thicknesses(in meters):16, 10, 10, 10, 10, 10, 13, 15, 20, 25, 30, 35, 40, 45, 50, 55, 60, 70, 80, 90, 100, 110, 120, 130, 140, 150, 170, 180, 190, 200, 220, 250, 270, 300, 350, 400, 450, 500, 500, 600. The model is driven by atmospheric data from a CMIP5 20th century simulation with the MPI-ESM LR setup.*

The authors describe the setup of simulations performed, but a description of the observations used (WOA13, SeaWiFS and MODIS-Aqua) is missing. I suggest including a brief description of these datasets.

*We have provided more accurate references to the data used, corrected the error (we are using MODIS Terra data, not MODIS Aqua), added a short description of the data after L.155 in the following form:*
*We compare the calculation results for both experiments (INDJ and INDB) with the best observational datasets to date for the region under consideration: they include oceanographic data compiled in the World Ocean Atlas 2013 (WOA13, Levitus et al., 2014) and the satellite data from the Sea-viewing Wide Field-of-view Sensor (SeaWiFS) and Moderate-resolution Imaging Spectroradiometer (MODIS) Terra. The WOA13 is a set of climatological mean, gridded fields of oceanographic variables based on in-situ measurements from a wide variety of sources. Global, decadal averages of temperature, salinity, oxygen and nutrients are provided at monthly, seasonal*

*and annual averaging periods on 102 standard depth levels from 0 to 5500m, and at 0.25 °
(temperature, salinity) and 1 ° (all variables) horizontal resolutions. We do not use the latest
edition of WOA18 (Boyer et al., 2018) for the following reasons: 1) WOA13 is widely used in the
scientific literature and thus can be easily compared to other studies that use the same reference,
2) as stated in the WOA18 description
([https://www.ncei.noaa.gov/data/oceans/woa/WOA18/DOC/woa18documentation.pdf](https://www.ncei.noaa.gov/data/oceans/woa/WOA18/DOC/woa18documentation.pdf)) WOA18
temperature and salinity data is still published as preliminary in order to take advantage of
community-wide quality assurance and comments.*

*From satellite data, we use SeaWiFS chlorophyll data (NASA Goddard Space Flight Center ...,
2021a) and MODIS Terra chlorophyll data (NASA Goddard Space Flight Center ..., 2021b), as
well as SeaWiFS downwelling diffuse attenuation coefficient data (NASA Goddard Space Flight
Center ..., 2021c). The SeaWiFS instrument was launched on the OrbView-2 satellite in August
1997, and collected data from September 1997 until the end of mission in December 2010. MODIS
is a key instrument aboard the Terra (EOS AM) and Aqua (EOS PM) satellites and its set of data
records covers the period from 24 February 2000 to present time. From above satellite data we
used their gridded fields of 9km resolution having daily and monthly averaging periods.*

*References:*

*Levitus, Sydney; Boyer, Tim P.; Garcia, Hernan E.; Locarnini, Ricardo A.; Zweng, Melissa M.;
Mishonov, Alexey V.; Reagan, James R.; Antonov, John I.; Baranova, Olga K.; Biddle, Mathew;
Hamilton, Melanie; Johnson, Daphne R.; Paver, Christopher R.; Seidov, Dan (2014). World
Ocean Atlas 2013 (NCEI Accession 0114815). NOAA National Centers for Environmental
Information. Dataset. [https://doi.org/10.7289/v5f769gt](https://doi.org/10.7289/v5f769gt) Accessed on 30/11/2021.*

*Boyer, Tim P.; Garcia, Hernan E.; Locarnini, Ricardo A.; Zweng, Melissa M.; Mishonov, Alexey
V.; Reagan, James R.; Weathers, Katharine A.; Baranova, Olga K.; Seidov, Dan; Smolyar, Igor
V. (2018). World Ocean Atlas 2018. NOAA National Centers for Environmental Information.
Dataset. https://accession.nodc.noaa.gov/NCEI-WOA18. Accessed on 30/11/2021.*

*NASA Goddard Space Flight Center, Ocean Ecology Laboratory, Ocean Biology Processing
Group. (2021a). Sea-viewing Wide Field-of-view Sensor (SeaWiFS) Chlorophyll Data; 2018
Reprocessing. NASA OB.DAAC, Greenbelt, MD, USA. doi:
10.5067/ORBVIEW-2/SEAWIFS/L3M/CHL/2018. Accessed on 01/05/2021*

*NASA Goddard Space Flight Center, Ocean Ecology Laboratory, Ocean Biology Processing
Group. (2021b). Moderate-resolution Imaging Spectroradiometer (MODIS) Terra Chlorophyll
Data; 2018 Reprocessing. NASA OB.DAAC, Greenbelt, MD, USA. doi:
10.5067/TERRA/MODIS/L3B/CHL/2018. Accessed on 01/05/2021*

*NASA Goddard Space Flight Center, Ocean Ecology Laboratory, Ocean Biology Processing
Group. (2021c). Sea-viewing Wide Field-of-view Sensor (SeaWiFS) Downwelling Diffuse
Attenuation Coefficient Data; 2018 Reprocessing. NASA OB.DAAC, Greenbelt, MD, USA. doi:
10.5067/ORBVIEW-2/SEAWIFS/L3M/KD/2018. Accessed on 01/05/2021*

L166: Why did you not use a more recent version of the World Ocean Atlas?

*We use WOA13 because:*

*1. WOA13 is widely used in the scientific literature and thus can be easily compared to other studies that use the same reference.*
*2. As stated in the WOA18 description (https://www.ncei.noaa.gov/data/oceans/woa/WOA18/DOC/woa18documentation.pdf) WOA18 temperature and salinity data is still published as preliminary in order to take advantage of community-wide quality assurance and comments. Therefore, we would avoid confusion with future studies using the WOA18 final version.*

*We have inserted this explanation into the text (see answer to the previous remark)*

L173: Since you use the monsoon season defined as June-September (JJAS), I suggest
changing "summer season" to "monsoon season" or "extended summer" to be coherent with the seasonal periods defined in L169-172. In L180-181, the pre-monsoon and monsoon seasons are referred. Following your first definition (L169-172), this means MAM
and JJAS. Please, check all the text and change accordingly with your first definition, also to be coherent with Section 4 where the summer season is defined as JJA.

*As recommended by the reviewer, we have changed the names of the seasonal periods throughout the text to keep the names defined in L169-172. Now the summer season in Section 4 is defined as JJAS, that is, in the same way as in other sections of the article. Figure 16 is replotted in accordance with the used definition of seasons.*

L201-203: The spatial distribution of the difference between the INDB and WOA13 SST, SSS and NO3 are identical to Fig2 for INDJ? The authors do not show these results but mention here the INDB simulation.

*The spatial distributions of SST, SSS, and other fields in the INDJ and INDB experiments are close to each other, but they are not identical. The difference is related to the impact of marine biogeochemistry's feedback. We discuss the differences between them in detail (for SST, SSS, primary production and nitrates) in section 3.1.4 of the manuscript.*
*To avoid confusion in L201-203 highlighted by the Reviewer, we have clarified this sentence:*
*The maximum deviations in surface nitrate field between the model and WOA13 data occur during the bloom periods (winter and monsoon seasons), while this deviation is minimal in pre-monsoon season.*

L222-224: Did you perform the same setup but with lateral boundary conditions driven by a reanalysis dataset?

*Yes, we did. But it was mainly focused on regional downscaling problems. I.e. on investigation on so-called "added value", etc. The results can be found in Mishra et al. https://epic.awi.de/id/eprint/54121/1/s2_0_S0169809521002337_main.pdf*

L255: Why the analysis period is different in Figure 4? Is it related with the availability of satellite data?

*Yes, the Reviewer is correct, the period of analysis presented in Fig. 4 is related to the availability of MODIS Terra (2000-2021) and SeaWiFS (1997-2010) data. Because our simulations span up to 2005, the resulting common period for the model results and satellite data is 1997-2005, which is used in Fig. 4 to calculate mean values of surface chlorophyll-a concentration for this period.*

*This text has been added to the manuscript to clarify the choice of this period:*

*The period of analysis presented in Fig. 4 is related to the availability of MODIS Terra (2000-2021) and SeaWiFS (1997-2010) data. Because our simulations span up to 2005, the resulting common period for the model results and satellite data is 1997-2005, which is used in Fig. 4 to calculate mean values of surface chlorophyll-a concentration for this period.*

L376: Why was the WOA2001 included? In L384-403, there is no discussion about the added value of including two WOA datasets.

*The reviewer is absolutely right: including the old WOA dataset to our comparison cannot add anything new. Therefore we removed the old release WOA2001 from the Figure 13 and corrected the relevant text.*

Technical corrections:
L190: "Sea surface concentration of dissolved nitrate." -> Italic style

*Corrected*

L368: "Thermocline dynamics." -> Italic style

*Corrected*

L500: "Fig. 16" or "Figure 16"

*Corrected*